# High Resolution Land Surface Modelling over Africa: the role of uncertain soil properties in combination with forcing temporal resolution.

Bamidele Oloruntoba[1,2], Stefan Kollet[1,2], Carsten Montzka[1], Harry Vereecken[1,2], Harrie-Jan Hendricks Franssen[1,2]

[1] Forschungszentrum Jülich, Institute of Bio- and Geosciences: Agrosphere (IBG-3), 52425 Jülich, Germany
[2] Centre for High-Performance Scientific Computing in Terrestrial Systems, Geoverbund ABC/J, 52425 Jülich, Germany.

*Correspondence to*: Bamidele Oloruntoba (b.oloruntoba@fz-juelich.de)

**Abstract.** Land surface modelling runs with CLM5 over Africa at 3km resolution were carried out and we assessed the impact of different sources of soil information and different upscaling strategies of the soil information, also in combination with different atmospheric forcings and different temporal resolutions of those atmospheric forcings. FAO and SoilGrids250m were used as soil information. SoilGrids information at 250m resolution was upscaled to the 3km grid scale by three different methods: (i) random selection of one of the small SoilGrids250m grid cells contained in the model grid cell; (ii) arithmetic averaging of SoilGrids soil texture values and (iii) selection of the dominant soil texture. These different soil model inputs were combined with different atmospheric forcing model inputs, which provide inputs at different temporal resolutions: CRUNCEPv7 (6-hourly input resolution), GSWPv3 (3-hourly) and WFDE5 (hourly). We found that varying the atmospheric forcing influenced simulated states and fluxes by CLM5 much more than changing soil information. Varying the source of soil texture information (FAO or SoilGrids250m) influences model water balance outputs more than the upscaling methodology of the soil texture maps. However, for high temporal resolution of atmospheric forcings (WFDE5) the different soil texture upscaling methods result in considerable differences in simulated evapotranspiration (ET), surface runoff and subsurface runoff at the local and regional scales related to the higher temporal resolution representation of rainfall intensity in the model. The upscaling methodology of fine scale soil texture information influences land surface model simulation results, but only clearly in combination with high temporal resolution atmospheric forcings.

## 1. Introduction

Understanding the intricate dynamics of land surface models (LSMs) over Africa involves a detailed examination of soil properties, which are indispensable yet steeped with uncertainty. The heterogeneity and complexity of soil properties (Vågen et al., 2016; Hengl et al., 2021) influence LSM simulations (Li et al., 2022), yet they often remain inadequately described within LSMs (Xu et al., 2023) due to limited data availability as a result of spatially

insufficient measurements (Dube et al., 2023). This inadequacy is further exacerbated in LSMs by the need to represent the point scale measurements at a coarse spatial resolution for field, regional or continental scale studies. Consequently, upscaling of soil information becomes a critical undertaking, aiming to bridge the gap between the fine-scale variability of soil properties and the broader scale at which LSMs usually operate (Van Looy et al., 2017; Montzka et al., 2017).

Quality of input datasets, like atmospheric forcings, soil physical properties or land surface parameters were found to greatly impact land surface modelling. Vahmani & Hogue (2014) compared remotely sensed green vegetation fraction (GVF) and impervious surface area (ISA) with the default look-up table derived values of the same parameters. The authors found that using the remotely sensed parameters, the model was able to replicate the observed ET. The feat was attributed to capturing all year-round irrigation by the remotely sensed data in the domain of interest. This highlights the importance of the source of input datasets into LSMs. The sensitivity of land surface models to atmospheric forcings as exemplified by Traore et al. (2014) over Africa was analyzed with two atmospheric forcing datasets; Watch Forcing Data Era Interim (WFDEI) and Watch Forcing Data (WFD). These two reanalysis datasets were generated using the same methodologies but with a slight difference in their source datasets (Weedon et al., 2014). The results showed that although there is a poor performance of ET in Central African forests, WFDEI was closer to eddy covariance measurements than WFD with correlations between 0.25 and 0.40. Lovat et al. (2019) used the ISBA-TOP coupled system (Bouilloud et al., 2010) over locations in the Mediterranean region at varying resolutions to assess river discharge and spatial runoff. It was noted that soil texture influences river discharge and runoff more than land cover does. Tafasca et al. (2019) used the land surface model ORCHIDEE and various global soil texture maps and noted that SoilGrids1km upscaled to 0.5º by selecting the dominant soil type generated similar water budgets as the 5 arc-min FAO Soil Map of The World (Reynolds et al., 2000) and the 1º resolution Global Soil Types map of Zobler (1986). The authors however indicated that the weak model sensitivity to the soil texture variation could have been caused by the coarse spatial resolution of 0.5º at which soil texture was discretized in the ORCHIDEE model.

These existing gaps in data and the critical impact of their uncertainties on LSM performance highlight the need for detailed studies. Studying how varying resolutions of soil and atmospheric data affect high resolution LSM outputs across diverse African ecosystems could help in refining model parameters and improving prediction accuracy. Furthermore, exploring new methods for effectively upscaling fine-scale soil measurements to broader applications in LSMs could provide insights into more robust upscaling strategies.

In this work, we are concerned with understanding the role of high-resolution soil texture input (at 3km horizontal spatial resolution) and its upscaling in the Community Land Model version 5.0 (Lawrence et al., 2019) (hereafter, CLM5) simulations over the entire African continent. This study investigates the impact of uncertainty in soil input variables and the upscaling method of soil texture information, at different spatial scales (from local to continental), and also in combination with different temporal resolutions of atmospheric forcings. The aim is not to compare simulations with measurements, but to detect model-internal sensitivities to (the upscaling of) soil texture information.

Twelve simulations combining four different soil texture inputs (FAO (Global Soil Data Task, 2014) and three differently upscaled SoilGrids maps (Hengl et al., 2017)) and three different meteorological forcings were

carried out and results are analysed in this work at the continental, regional and point scale. We also compared our outputs with an external dataset, GLDAS-2.1 (Rodell, 2020) to assess the performance of the different upscaling methods. The novelty of this work lies in the detection of the impact of uncertainty in the (upscaling of) soil texture information, especially in combination with different temporal resolutions of atmospheric forcings. The impact of uncertainties of atmospheric forcings on land surface model simulations over Africa has been studied (Boone et al., 2009 and Iyakaremye et al., 2021) but its interaction with the uncertainties in soil information has not been studied over Africa at a high spatial resolution.

This research therefore seeks to answer the following questions: 1) Are simulation results of CLM5 sensitive to different soil texture inputs, and different upscaling methods applied to soil texture input? 2) What is the role of the temporal resolution of atmospheric forcings in combination with the different soil texture inputs?

## 2. Materials and Methods

### 2.1. CLM5

CLM5 is a mechanistic land surface model which represents land surface heterogeneity differently from most other land surface models previously used over Africa in continental simulations (Traore et al., 2014; Ghent et al., 2010a ; Weber et al., 2009). While some of the models previously used over Africa had a single layered sub-grid system popularly known as mosaic system, CLM5 uses a multi-layered sub-grid hierarchy. This means that in CLM5, each grid cell represents multiple land units consisting of vegetated, lake, urban and glacier areas. Each land unit represents multiple columns which could have different soil profiles with autonomously evolving vertical profiles of soil moisture content and temperature, and each column has multiple patches of Plant Functional Type (PFT) or Crop Functional Type (CFT) (Lawrence et al., 2018). Among the numerous improvements of CLM5 compared to its predecessor, are the inclusion of a spatially variable soil depth, replacement of Ball-Berry by Medlyn stomatal conductance and updated irrigation scheduling (Lawrence et al., 2019). Considering Africa's land surface heterogeneity, CLM5 has features of great interest for land surface modelling over Africa at a high spatial resolution.

CLM5 provides a framework for modelling the soil processes necessary for understanding terrestrial hydrology. This version of the model improves the representation of soil porosity and pore size distributions, considering both mineral and organic components of the soil (Lawrence et al., 2018).Saturated hydraulic conductivity and soil matric potential are calculated using Cosby et al. (1984) for mineral soils, with modifications to accommodate the effects of organic matter based on its depth and content. Detailed equations are comprehensively provided in Lawrence et al. (2018).

Furthermore, CLM5 employs the Brooks and Corey model (Brooks and Corey, 1964) to associate soil moisture content with water potential, considering the variability in soil texture using organic and mineral soil fraction parameters.

$$B_i = \left(1 - f_{\{om,i\}}\right) \cdot B_{\{min,i\}} + f_{\{om,i\}} \cdot B_{\{om\}} \qquad (1)$$

where $B_{\{om\}}$ covers organic matter, $f_{\{om,i\}}$ represents organic matter fraction and

$$B_{\{min,i\}} = 2.91 + 0.159 \cdot (\%\{clay\})_i \qquad (2)$$

is for mineral soil where $(\%\{clay\})$ is the percentage of clay in each grid cell at level i.

The full water balance equation now implies interactions between canopy, surface, soil, and aquifer water and ice storage, defining the model's detailed approach to hydrologic partitioning over varying temporal scales.

The water balance equation is represented by:

$$\Delta Sc,l + \Delta Sc,sn + \Delta Ssfc + \Delta Ssn + \sum_{i=1}^{N_{levsoi}}\left(\theta s_{liq,i} + \theta s_{ice,i}\right) + \Delta Sacq =$$
$$(qrn + qsn - Ev - Eg - qover - qsfcwat - qdr - qrgl - qsnsfc)\,\Delta t \qquad (3)$$

where $\Delta Sc,l$ represents changes in canopy water, $\Delta Sc,sn$ changes in canopy snow, $\Delta Ssfc$ changes in surface water, $\Delta Ssn$ changes in surface snow and $\Delta Sacq$ changes in water stored in the aquifer. $\theta s_{liq,i}$ represents changes in soil water, $\theta s_{ice,i}$ represents changes in soil ice at each soil level i. $N_{levsoi}$ refers to the number of soil levels. On the right-hand side of the equation, qrn represents rainfall, qsn snowfall, Ev transpiration, Eg evaporation, while qover refers to surface runoff, qsfcwat runoff from surface water storage, qdr drainage, qrgl glacier and lakes runoff and qsnsfc snow-capped surface runoff. Precipitation (qrn + qsn) is intercepted by canopy, which is controlled by leaf area index. The moisture input reaching the surface after evaporative losses from both the vegetation and surface (Ev , Eg) is then divided between surface runoff, surface water storage and infiltration. The units for fluxes are kg/m$^2$s while storage variables are quantified in kg/m$^2$ and $\Delta t$ is in mm/second. For a detailed description of the mathematical formulations and their applications within CLM5, readers are referred to Lawrence et al. (2018), where these processes are described in-depth.

Irrigation in CLM5 separates irrigated and rainfed crops by assigning them to separate soil columns. Irrigation is applied daily at 6am based on the difference between soil moisture content and target soil moisture taking also the crop leaf area index into account. Irrigation decisions are guided by datasets detailing areas equipped for irrigation according to Portmann et al. (2010). To constrain CLM5 irrigation, irrigation water is sourced from river storage, with provisions for supplements from ocean reserves. Alternatively, in severe cases of water scarcity, irrigation demand is dynamically adjusted to conserve river water levels. The applied irrigation in CLM5 is hard coded to bypass canopy interception, meaning it is added directly to the ground surface. More details can be found in Lawrence et al. (2018).

## 2.2. Soil Texture Information

Soil hydraulic and thermal properties are critical for flux and state calculations in LSMs (Zhao et al., 2018). These values are generally obtained from soil texture information through pedotransfer functions, which is also the case for CLM5. Two different soil texture datasets, the IGBP-DIS Soil Dataset and SoilGrids250m dataset, were used as input for CLM5 simulations over Africa. The IGBP-DIS soil dataset was generated using the linkage method which is characterized by lack of intra-polygonal variation. This soil texture dataset is the default soil texture information available in CLM5. The soil texture dataset which is at approximately 8km resolution provides information for the top 10 CLM5 soil layers: at 0.0175, 0.0451, 0.0906, 0.1656, 0.2892, 0.493, 0.829, 1.3829, 2.2962 and 3.4332 meter depth.

ISRIC's SoilGrids250m (Hengl et al., 2017) was produced by machine learning and it is the successor of the SoilGrids1km product (Hengl et al., 2014). SoilGrids250m has a spatial resolution of 250m and is therefore considered because of its potential to better represent local scale soil processes related to the higher spatial resolution. When evaluated with soil profiles from WoSIS (World Soil Information Service), SoilGrids250m has a higher accuracy than FAO with a RMSE of 18.6% versus 26.3% for the sand fraction at 0-30cm depth and 12.5% versus 15.4% for clay fractions at this depth (Dai et al., 2019).

Improvements in SoilGids250m compared to its earlier version SoilGrids1km include for example further soil information for deserts and arid areas such as the Sahara Desert covering about 30% of Africa's land mass (Tucker & Nicholson, 1999). About 150,000 soil profiles were obtained globally across all continents from both actual and pseudo-observations. Actual observations were from in situ and remote sensing measurements and values reported by national classification systems. Pseudo-observations came from expert assessment of both restricted areas and places with extreme climate conditions like deserts, glaciers, mountain tops, tropical forests, and austere regions. SoilGrids250m provides global estimates for soil texture fractions, organic carbon, bulk density, cation exchange capacity, pH and coarse fragments. Compared to SoilGrids1km, SoilGrids250m records in sand, silt and clay contents, over 60% relative improvement as explained by a 10-fold cross validation exercise. The Soilgrids250m unlike the IGBP-DIS was provided at seven standard soil depths of 0, 0.05, 0.15, 0.30, 0.60, 1.00 and 2.00 meters depth.

**2.3. Upscaling of Soil Textural properties**

Upscaling of soil hydraulic properties is needed when the model resolution is coarser than the resolution of the measurement based product. SoilGrids250m soil texture information needs to be upscaled to the 3km x 3km resolution of the CLM5 model for Africa. One CLM5 grid cell contains therefore 144 SoilGrids250m grid cells. Three upscaling methods of soil texture information were compared in this work:

(i) Simple averaging of the soil texture values for all the SoilGrids grid cells which are contained in a larger CLM grid cell (e.g., Kochendorfer and Ramírez, 2010). Since both clay and sand soil texture information were provided as fractions per grid cell, a simple averaging of the fractions was performed.

(ii) Selection of the dominant soil type (according to USDA soil classification) in a CLM grid cell and use of the soil texture values for that soil type for the complete CLM grid cell. This method was for example used in Tafasca et al. (2019). The dominant soil type is any soil type with the highest representation among the 144 SoilGrids grid cells.

(iii) Random selection of a single SoilGrid cell and use of the soil texture values for this grid cell for the complete 3km x 3km CLM model grid cell. This method which is a novelty of this work, creates a chance for texture outliers to define the soil hydraulic parameters. This ensures that over larger regions the Probability Density Function (PDF) of soil properties is better reproduced by the model than by selecting the dominant soil texture or average soil texture. It differs from other upscaling methods as it avoids spatial averaging or smoothing. Although it can introduce larger local biases in the soil hydraulic parameters and thus model output variables, it is not expected to induce systematic

biases at larger scales, as local biases for some grid cells will be cancelled out by biases at other grid cells. In addition, as soil texture is not averaged or smoothed before processing it through the non-linear simulation model, it is expected that also model output variables, averaged over larger areas, are unbiased. We also specified a random number generator (RNG) seed which makes the randomisation reproducible in other machines.

**2.4. Meteorological Forcing datasets and evaluation dataset**

In this work, the impact of three different meteorologic forcing datasets with different temporal resolution, in combination with the different soil texture input datasets, was investigated. We examined CRUNCEPv7 (Viovy, 2018), GSWP3 (Hyungjun, 2017) and WFDE5 (the bias corrected ERA5 dataset using WATCH Forcing Data methodology) (Cucchi et al., 2020). These three forcings have been selected because they possess all atmospheric variables CLM5 requires, have similar spatial resolution and, especially, their varying temporal resolution of 6 hours (CRUNCEP), 3 hours (GSWP) and 1 hour (WFDE5). The impact of the varying temporal resolution was studied in combination with the different soil texture inputs. GSWPv3 and CRUNCEPv7 have been used in the past already in combination with CLM4, CLM4.5 and CLM5 (Bonan et al., 2019). WFDE5 has been tested at 13 globally spread FLUXNET2015 locations. Cucchi et al. (2020) showed that WFDE5 has smaller mean absolute errors and larger correlations of variables like precipitation, global radiation, specific humidity, air temperature, and wind speed with observations than the WFDEI (Watch Forcing data ERA Interim) dataset which was used in Traore et al. (2014) over Africa. For comparison and assessment of the different upscaling methods performance, the GLDAS-2.1 dataset was used. The dataset has been used over Africa to train deep learning algorithms for modelling groundwater (Gaffoor et al., 2022), calculate drought recovery time (Hao et al., 2022) and asses the spatio-temporal patterns of drought in East Africa (Liu et al., 2022). Our choice of GLDAS-2.1 dataset is motivated by the fact that it provides soil moisture information from 0-200cm.

(i) **CRUNCEPv7.** CRUNCEPv7 dataset is a combination of CRU (Climate Research unit Time Series) 3.24 (Harris, 2013) and National Centre for Environmental Protection (NCEP) reanalysis (Kalnay et al., 1996). The data are available for the period between 1901 and 2016 with a horizontal resolution of $0.5^{\circ}$ and 6 hourly temporal resolution. Precipitation, cloudiness, temperature, and relative humidity were taken from CRU while wind speed, pressure and long wave radiation were obtained from NCEP.

(ii) **GWSP3**. The Global Soil Wetness Project version 3 dataset is a 3-hourly, $0.5^{\circ}$ horizontal resolution atmospheric forcing product. The data are available for the period between 1900 and 2014 and are based on NCEP's 20th century reanalysis project (Compo et al., 2011). Though the 20th century project dataset was published at $2^{\circ}$ horizontal resolution, the GSWP version 3 dataset was downscaled to $0.5^{\circ}$ horizontal resolution using a spectral nudging technique (Yoshimura and Kanamitsu, 2008). Four out of seven variables namely air temperature, precipitation, long and short wave radiation were bias corrected using Climate Research Unit's CRU Tsv3.21 (Harris, 2013), Global Precipitation Climatology Centre's GPCCv7 (Schneider et al., 2014) and surface radiation budget datasets (Lawrence et al., 2019). GSWP3 is the default forcing provided with the CLM5

Model (Lawrence et al., 2018). Since both GSWP3 and CRUNCEPv7 datasets were provided for use in CLM by the developers there was no additional processing needed to use these datasets in CLM5.

(iii) **WFDE5**. The WFDE5 dataset was created by using the WATer and global CHange (WATCH) Forcing Data methodology to process near surface 5th generation ECMWF (European Centre for Medium-range Weather Forecasts) ReAnalysis (ERA5) variables. WFDE5 was provided globally on a regular lonlat grid at 0.5º x 0.5º spatial resolution at hourly time steps. It has therefore the highest temporal resolution of the considered atmospheric forcing datasets in this study. WFDE5 correlates better with FLUXNET2015 datasets at each site than WFDEI (Traore et al. 2014). Another advantage WFDE5 has over the higher spatial resolution ERA5 data set is that the monthly precipitation totals were bias corrected using precipitation data from the Climate Research unit Time Series (CRU TS) and Global Precipitation Climatology Centre (GPCC). This is important as precipitation has a large impact on LSM simulations compared to other meteorological forcings (Bucchignani et al., 2016) over Africa.

**GLDAS-2.1**. GLDAS-2.1 dataset was used for verification purposes in this work. The Global Land Data Assimilation System was originally developed to absorb satellite- and ground-based observational data products, using advanced land surface modelling and data assimilation techniques, in order to generate fields of land surface states and fluxes (Rodell et al., 2004). The GLDAS-2.1 dataset, which was reprocessed in January 2020, delivers monthly 0.25-degree data produced by temporal averaging of 3-hourly simulations using the Noah Model 3.6 in LIS Version 7. The GLDAS-2.1 simulations were driven by NOAA/GDAS atmospheric fields, GPCP V1.3 precipitation data, and AGRMET radiation variables from March 2001 onward. Table 1 summarizes details regarding the different meteorological forcing and evaluation datasets used in this work.

Table 1. Main properties of the reanalysis datasets CRUNCEPv7, GSWPv3, WFDE5 and GLDAS-2.1 used in this work.

| Properties | CRUNCEPv7 | GSWP3 | WFDE5 | GLDAS-2.1 |
|---|---|---|---|---|
| Spatial resolution | 0.5º | 0.5º | 0.5º | 0.25º |
| Temporal resolution | 6 Hourly | 3 Hourly | 1 Hourly | Monthly |

## 2.5. Model Setup and analysis

CLM5 was run in this work in land only mode, i.e., instead of coupling CLM5 with an atmospheric model, atmospheric reanalysis datasets are used as external forcings to the land surface model. Atmospheric input to CLM5 includes precipitation, incoming shortwave radiation, air temperature, surface air pressure, specific

humidity, wind speed and incoming longwave radiation. These are available for CRUNCEP every 6 hours, GSWP every 3 hours and for WFDE5 hourly. But since model time step is 30 minutes, precipitation is divided equally over the different model time steps. For air temperature, surface air pressure, specific humidity and wind speed, all values are interpolated to model time steps using nearest neighbour algorithm. For solar radiation, cosine of the solar zenith angle is used to ensure a smoother diurnal cycle, while preserving the total radiation from the atmospheric input data.

Sixteen plant functional types were activated, alongside transient $CO_2$ and aerosol deposition rates. All twelve model simulations (Table 2) apply monthly leaf area index (LAI) as observed from satellite phenology. A spatially varying soil thickness dataset (Pelletier et al., 2016) with values ranging from 0.4m to 8.5m was also applied. The land cover description is based on 1km resolution Moderate Resolution Imaging Spectroradiometer (MODIS) products. Land Cover Type is from MCD12Q1 version 5 which provides annual land cover intervals between 2000 and 2015.

Twelve simulations (3 atmospheric forcings combined with 4 soil texture maps) were performed over the CORDEX Africa domain which covers longitude -24.64°W to 60.28°E and latitude 45.76°S to 42.24°N (results over African continent only). The horizontal resolution for all model simulations was approximately 0.027°, i.e. about 3 km. This discretization results in 10,033,920 grid cells. Simulation period was from the 1st of January 2011 to the 31st of December 2014 and results for the first two years were discarded (spin up year). Earlier works over the Southern Africa region including Crétat et al. (2012), Ratna et al. (2014) and Zhang et al. (2023) have employed 6 months or less spin-up times using different land surface models while Zheng et al. (2017) employed 1 year for spin-up with a predecessor of CLM5 over the Tibetan Plateau. We compared the simulated water balance components in this work with water balance components (evapotranspiration, surface runoff and soil water content) from a fresh simulation which had 11-years of spin up time and the results do not alter our initial conclusion in this study (S54-S56). Moreover, we evaluated the adequacy of the reference period employed in this study. The continental annual average of the deepest soil moisture layer was calculated, a trend line was fixed, and the statistical significance was calculated to determine whether the slope of the trend differed significantly from zero. The resulting p-value of 0.353 indicated that the trend in soil moisture over the three-year period was not statistically significant based on a 95% confidence interval (S57), suggesting that extending the study period will not alter the current outcome.

Although the model time step size was 30 minutes, most results are presented as monthly sums (at regional and local scales). For continental scale results, annual mean of evapotranspiration (ET), surface runoff, and subsurface runoff were computed as well as the seasonal mean of the weighted average of the top 2 meters soil moisture content. The weights for calculating weighted average of soil moisture content were defined according to the thickness of each soil layer in CLM5.

To further substantiate the role of soil texture input to CLM5, a new set of simulations was conducted. To ensure comparability with CRUNCEP (6 hourly) and GSWP (3 hourly), the hourly WFDE5 forcings were aggregated to 6 hours and 3 hours, respectively. The model was then run with the soil texture information. This was conducted

to identify discrepancies between the simulation outcomes of WFDE5 at hourly, 3-hour and 6-hour temporal resolutions. Furthermore, a comparison was made with the results obtained by CRUNCEP and GSWP. The results were also analysed at the monthly level, in addition to the regional and local time series.

A metric termed "average margin" was introduced to quantify the impact of temporal resolution of atmospheric forcings in combination with soil texture map variation. The four soil texture maps were considered each providing a unique output at every timestep within the time series. The average margin for a simulated variable for a certain atmospheric forcing/soil texture map combination at a given time step is denoted by M1(t), M2(t), M3(t) and M4(t). The difference in the maximum and minimum simulated value for the variable, between the soil texture maps at a given time step is then computed as:

$$D(t) = \max(M_1(t), M_2(t), M_3(t), M_4(t)) - \min(M_1(t), M_2(t), M_3(t), M_4(t)) \qquad (4)$$

and the average margin is given by:

$$A = \frac{1}{T} \sum_{t=1}^{T} D(t) \qquad (5)$$

where T represents the total number of time steps in the time series and t denotes time step.

A one-way analysis of variance (ANOVA) was conducted to ascertain whether the outputs of the four soil maps for each atmospheric forcing group exhibited significant variation. Firstly, the mean of the four soil map outputs was calculated, and the deviation of each map's output from the mean was obtained. The resulting deviations were subsequently expressed as percentages relative to the mean output, thus providing a normalised measure of the deviation for each soil map, which could then be compared with results for other atmospheric forcings. The data were subsequently transformed into a long format suitable for ANOVA, in which the percentage deviations for each soil map were compared. The dependent variables were the obtained percentage deviations, while the independent variables were the categorical variable defining the compared groups (FAO, dominant, mean and random). Subsequently, an analysis of variance (ANOVA) was conducted to ascertain whether there were statistically significant discrepancies between the models' percentage deviations. The results of the ANOVA analysis yielded a p-value statistic, which was used to determine the significance of the observed variations in soil texture map outputs at the 95% confidence interval. For further details on the ANOVA framework, we direct the reader to the works of Fisher (1925) and Brandt (2014).

Finally, we compared the different soil texture map outcomes with the GLDAS-2.1 dataset as a benchmark to compare CLM5 model outputs to an established external dataset. We compared ET, surface runoff and soil moisture content using the Pearsons correlation (Pearson and Henrici, 1997) to measure the strength of relationship between the datasets and Root Mean Square Error (RMSE). More details about RMSE and its proper use are described by Hodson, (2022). For the reference study period, for every grid cell and all time steps the calculated water balance components were compared with the ones from the GLDAS-2.1 dataset. This comparison was performed on a grid cell-by-grid cell basis, resulting in a complete continental assessment of the water balance components."

Table 2: Summary of CLM5 experiments in this study, combing different soil texture input information and atmospheric forcings.

| Experiment | Soil Texture | Forcing |
| --- | --- | --- |
| FAO_CRU | FAO | CRUNCEP |
| SGd_CRU | SoilGrids-Dominant | CRUNCEP |
| SGm_CRU | SoilGrids-Mean | CRUNCEP |
| SGr_CRU | SoilGrids-Random | CRUNCEP |
| FAO_GSW | FAO | GSWP |
| SGd_GSW | SoilGrids-Dominant | GSWP |
| SGm_GSW | SoilGrids-Mean | GSWP |
| SGr_GSW | SoilGrids-Random | GSWP |
| FAO_WFD | FAO | WFDE5 |
| SGd_WFD | SoilGrids-Dominant | WFDE5 |
| SGm_WFD | SoilGrids-Mean | WFDE5 |
| SGr_WFD | SoilGrids-Random | WFDE5 |
| FAO_WFD | FAO | WFDE5-3H |
| SGd_WFD | SoilGrids-Dominant | WFDE5-3H |
| SGr_WFD | SoilGrids-Random | WFDE5-3H |
| FAO_WFD | FAO | WFDE5-6H |
| SGd_WFD | SoilGrids-Dominant | WFDE5-6H |
| SGr_WFD | SoilGrids-Random | WFDE5-6H |

**2.6 Definition of regions**

Iturbide et al. (2020) updated the IPCC climate reference regions for subcontinental analysis based on, amongst
others, coherence of climate variables. The new reference regions for Africa include the Mediterranean, Sahara,
West Africa, North-East Africa, Central Africa, Central-East Africa, South-West Africa, and South-East Africa.
Here we combined South-East Africa and Madagascar into one region. Figure S1 shows the African sub-regions.
The eight regions are used as basis to calculate region-specific water balance components.

**3. Results and Discussion**

### 3.1. Comparison of simulated water balance components with GLDAS-2.1 Datasets

To assess the agreement between the CLM5-simulated water balance components and a reference dataset, a comparison was conducted with the outputs of GLDAS-2.1. We acknowledge that while the GLDAS-2.1 serves as a benchmark for comparison, the extent to which it accurately represents actual conditions remains uncertain.

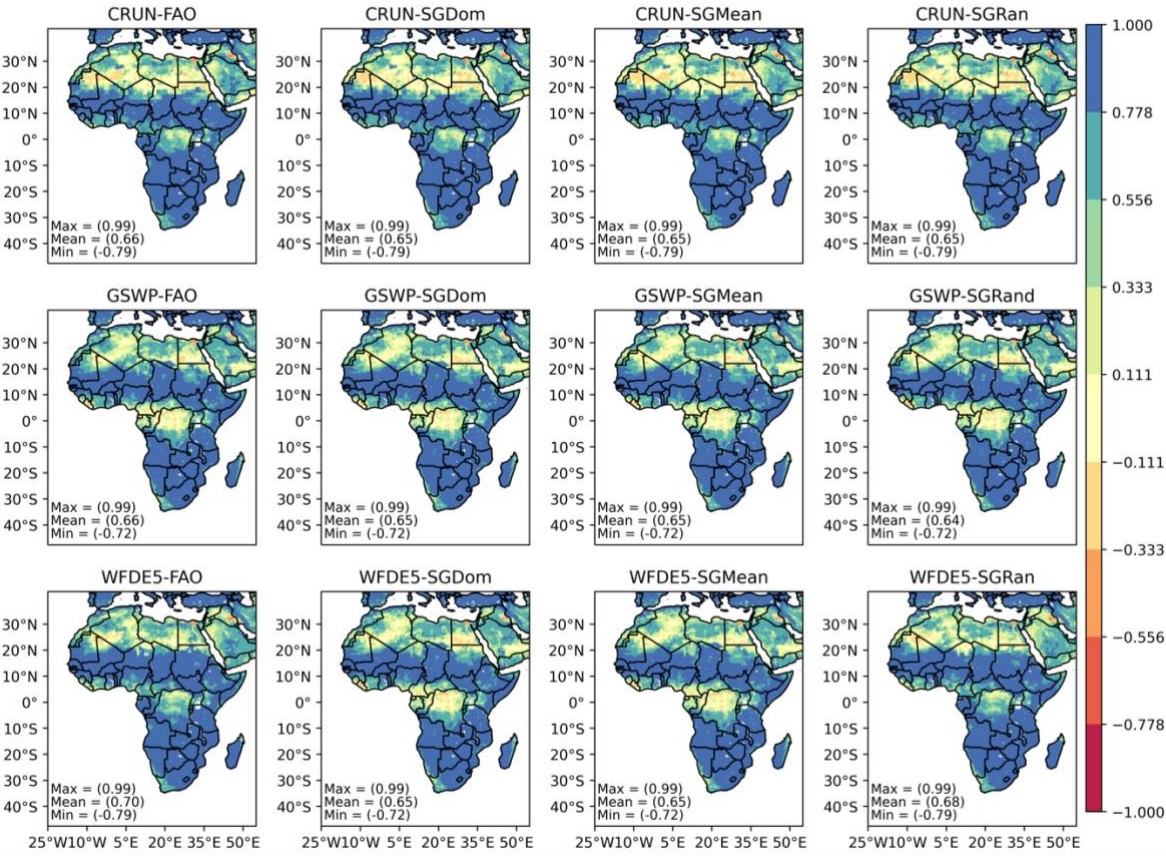


**Figure 1: Temporal correlation maps of simulated evapotranspiration compared with the Global Land Data Assimilation System (GLDAS-2.1) dataset over Africa for three different atmospheric forcing datasets (CRUN, GSWP, and WFDE5) and four soil texture maps (FAO, SGDom, SGMean, and SGRan). Top Row: Correlation maps for the CRUNCEP dataset using the FAO, SGDom, SGMean, and SGRan soil texture maps. Middle Row:**

**Correlation maps for the GSWP dataset using the same four soil texture maps. Bottom Row: Correlation maps for the WFDE5 dataset using similar maps.**

**Evapotranspiration**

The correlation of CLM5 simulated ET with GLDAS (Figure 1) shows a clear spatial gradient across Africa. Strong positive correlations above 0.75 as referenced in hydrology studies over Africa (Scanlon et al., 2022; Larbi

et al., 2020) are mainly seen in the equatorial region and parts of Eastern Africa, Southern Africa and Madagascar, indicating acceptable model performance in these regions. Northern Africa, some parts of Central Africa, and the cape of South Africa tend to show moderate to weak positive correlations, with some areas having negative correlation (down to around -0.79). The mean correlation values hover around 0.64-0.70, reflecting relatively moderate agreement with GLDAS across the continent. RMSE for ET (Figure S50) displays a concentration of

lower errors in the moisture deficient Northern and Southern parts of Africa, while the moisture richer Central

and Eastern regions show higher RMSE values. This suggests that while CLM5 simulated ET corresponds well with GLDAS in the equatorial zones, there is higher variability and model uncertainty in the arid and semi-arid regions. It is important to note however that RMSE scores are magnitude dependent as they increase or decrease with the magnitude of evaluated variables.


**Surface Runoff**

Surface runoff correlations (Figure 2) over Africa exhibit wide variability, with very high positive correlations (up to 1.0) in Savannah regions of West Africa including parts of Namibia, Zambia and Mozambique. There are however areas with low to strongly negative correlations, particularly in Mauritania, Mali, Algeria, Libya, Egypt

and Sudan, where correlation values are as low as -1.0. This high variability results in an average continental correlation of 0.50-0.58. The RMSE for surface runoff over Africa (Figure S52) shows minimal errors in water scarce Northern and South-western Africa, with the highest RMSE values ranging from 0-11mm/month . Central Africa and Western regions show relatively higher RMSE values. The high RMSE values suggest substantial discrepancies in surface runoff simulation between CLM5 and GLDAS, especially in equatorial areas.

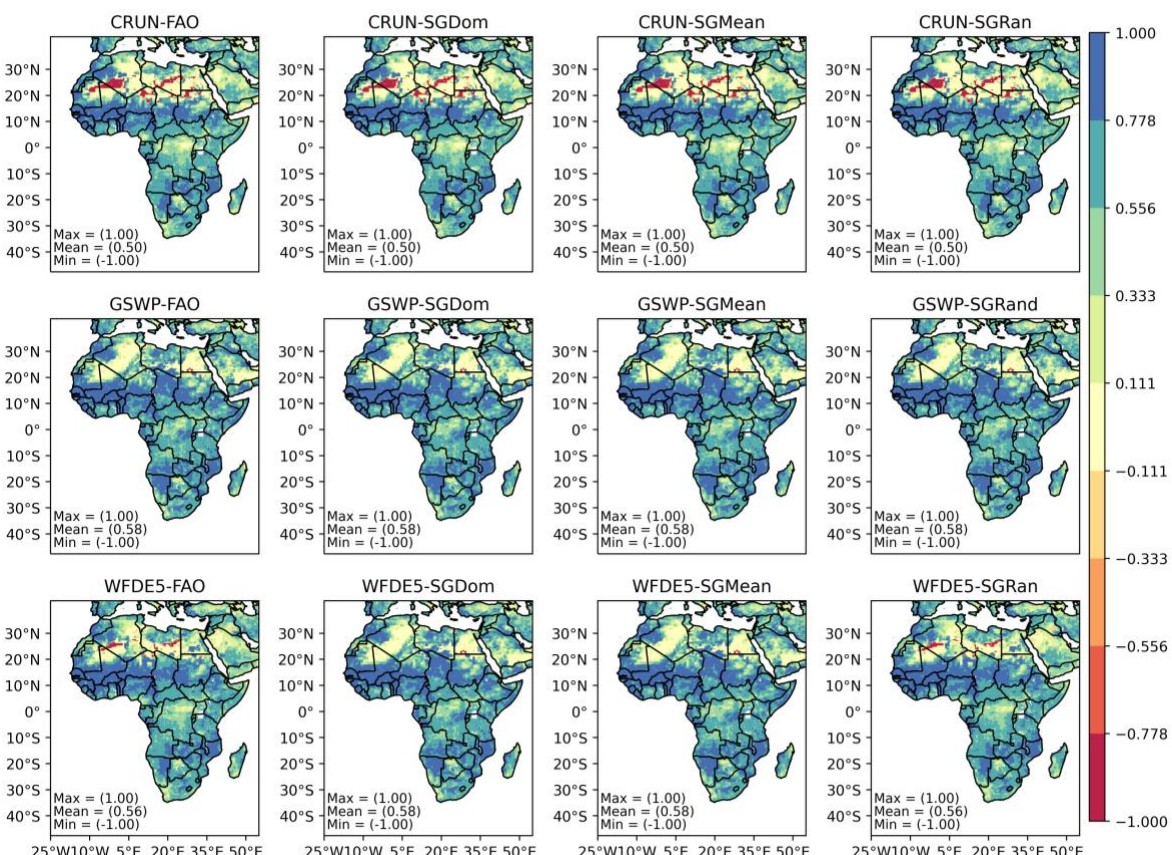


**Figure 3: Temporal correlation maps of simulated surface runoff compared with the Global Land Data Assimilation System (GLDAS-2.1) dataset over Africa for three different atmospheric forcing datasets (CRUN, GSWP, and WFDE5) and four soil texture maps (FAO, SGDom, SGMean, and SGRan). Top Row: Correlation maps for the CRUNCEP dataset using the FAO, SGDom, SGMean, and SGRan soil texture maps. Middle Row: Correlation maps**

**for the GSWP dataset using the same four soil texture maps. Bottom Row: Correlation maps for the WFDE5 dataset using similar maps.**

### Soil Moisture

Soil moisture correlations with GLDAS (Figure 3) show a slightly different spatial pattern compared to ET. The

highest correlations (strong positive) are generally observed above the equator, top fringes of Southern Africa and Northern Madagascar. Strong negative correlations however are found in parts of Sahara specifically Mauritania, Mali, Algeria, Egypt and Sudan where certain grid cells exhibit correlations as low as -0.79. Overall, the average correlations for soil moisture are lower than for ET with a range of 0.56-0.67, indicating less correlation across the continent compared to ET. The RMSE map for soil moisture (Figure S51) exhibits average values ranging

between 0.05-0.06 cm$^3$/cm$^3$. There is slightly higher RMSE in parts of Central and Africa specifically in Congo DR, where errors peak around 0.26-0.27 cm$^3$/cm$^3$. This RMSE pattern suggests that the CLM5 simulated soil moisture maintains a relatively stable agreement with GLDAS having minimal extreme errors across the continent.

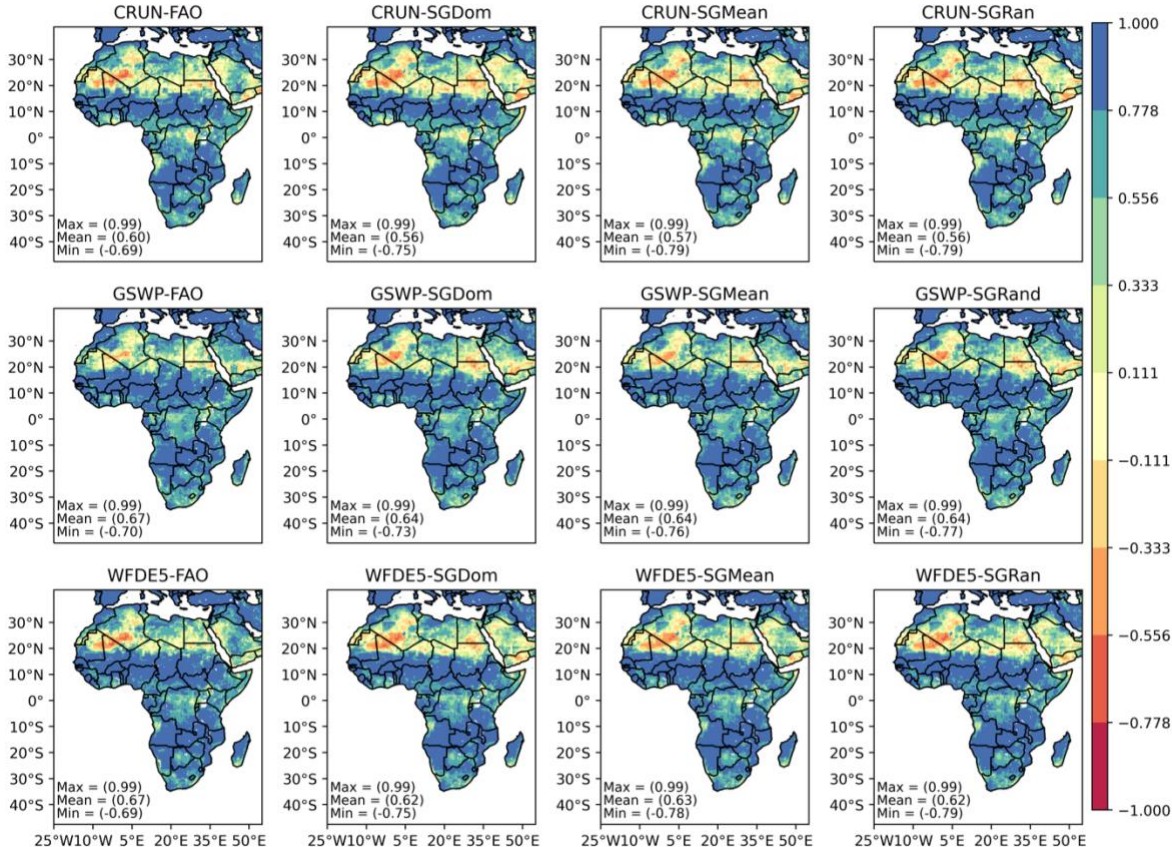

**Figure 2: Pearson correlation maps of simulated soil water content compared with the Global Land Data Assimilation System (GLDAS-2.1) dataset over Africa for three different atmospheric forcing datasets (CRUN, GSWP, and WFDE5) and four soil texture maps (FAO, SGDom, SGMean, and SGRan). Top Row: Correlation maps for the CRUNCEP dataset using the FAO, SGDom, SGMean, and SGRan soil texture maps. Middle Row:**


## 3.2. Continental Simulated Water Balance Components

### 3.2.1 Evapotranspiration

Figure 4 shows actual ET estimates over Africa for the different soil texture maps used in this study and the different atmospheric forcings. Continental average ET and local ET maxima were estimated for all 12 simulations for the reference period of 2013-2014.

The soil texture map has in general only a limited impact on simulated ET. For CRUNCEP forced simulations the yearly ET varies among the soil maps between 452.9 mm/year and 454.4 mm/year, with the lowest ET for the

FAO soil texture map and slightly higher ET for the SoilGrids texture maps. Also, for GSWP forced simulations we find the lowest simulated ET for the FAO soil texture map (438.7 mm/year), while the highest simulated ET is only slightly higher (439.6 mm/year). Simulated ET is highest for the SoilGrids soil map which is randomly upscaled. Also, for the WFDE5 simulations differences in simulated ET are very small and vary between 442.5 mm/year (FAO) and 443.5 mm/year (SoilGrids, randomly upscaled). These numbers also illustrate that the impact

of variations in soil texture input are much smaller than variations in atmospheric forcings. While the four different soil texture maps result in maximum variations in average yearly ET over the African continent of only ~1mm for a given atmospheric forcing, the variations in atmospheric forcings result in maximum variations in average yearly ET over the African continent around 14mm/year, for a given soil texture dataset. Specifically, the upscaling procedure of the soil texture information exhibits negligible effects on the mean annual estimates of

evapotranspiration over Africa. Also, the maximum simulated ET for a grid cell over Africa is hardly affected by the soil texture map input (<1mm/year), with even smaller variations among soil texture maps than the continental average. On the other hand, variations in atmospheric forcings affect the local maximum simulated yearly ET stronger with variations among forcings ~36mm/year.

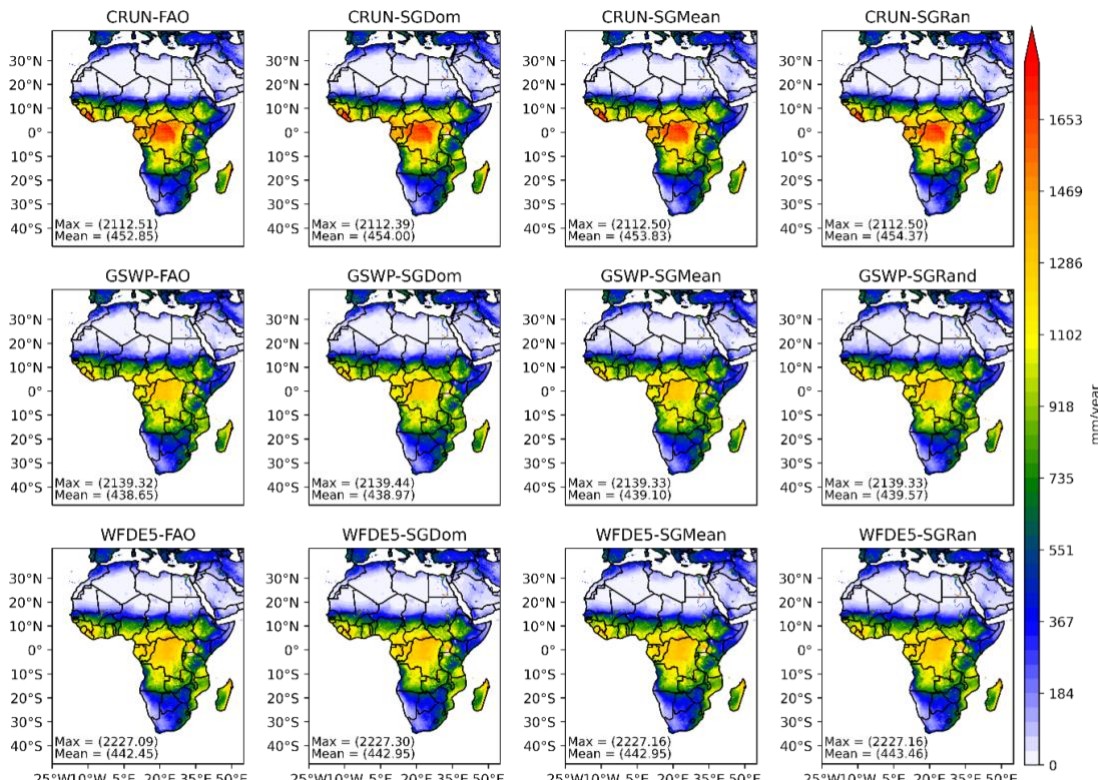

**Figure 4: Spatial distribution of simulated mean annual evapotranspiration over Africa. Upper row: CRUNCEP forced simulations with, from left to right, FAO, Dominant, Mean and Random upscaled soil texture map inputs. Middle row: like row 1, but GSWP forced simulations. Bottom row: similar to row 1, but WFDE5 forced simulations.**

### 3.2.2 Surface runoff

Also, the continental surface runoff is not strongly affected by variations in the soil texture map (Figure 5). For all three atmospheric forcings, the average surface runoff over the African continent is almost the same for the four different soil texture maps and differences in surface runoff are never larger than 0.3mm/year, for a given atmospheric forcing. When examining the influence of soil texture maps on surface runoff, it becomes evident that the disparities between the various SoilGrids maps, generated using different upscaling methods, are minimal. The maximum difference in continental averages of surface runoff between the SoilGrids soil texture maps with the highest and lowest values is only 0.01-0.02 mm/year, depending on the atmospheric forcing. However, slightly larger differences are observed when comparing the FAO soil texture map with the SoilGrids texture maps, with a maximum variation of 0.20-0.26 mm/year, again depending on the atmospheric forcing. These findings indicate that while the upscaling process of soil texture maps does not substantially impact simulated surface runoff with CLM5, the source and type of soil texture maps employed do have a small, yet perceivable, influence on the results.

On the other hand, the atmospheric forcing shows a much larger impact on average surface runoff over Africa with a value of approximately 94 mm/year for CRUNCEP (6 hourly temporal resolution), 114 mm/year for GSWP (3-hourly temporal resolution of atmospheric forcings) and 122 mm/year for WFDE5 (hourly forcings). Spatial details can be found in Figure 2. The substantial difference of 28 mm/year in average annual surface runoff

between WFDE5 and CRUNCEP contributes potentially to higher ET estimates for CRUNCEP by 11 mm/year. The increased surface runoff in the WFDE5 forced simulations reduces the availability of water for ET processes especially after runoff events.

The differences in surface runoff could be related to the temporal resolution of the atmospheric forcings. A higher temporal resolution of the atmospheric forcings as for WFDE5 will result in higher peaks of precipitation intensity, whereas a coarser temporal resolution of 6 hours like for CRUNCEP will average out intensive precipitation over longer time periods with less high peaks in precipitation intensity. As surface runoff is generated under conditions of (very) high precipitation intensity, it can be expected that the temporal resolution of the atmospheric forcings

will affect the simulated amount of surface runoff.

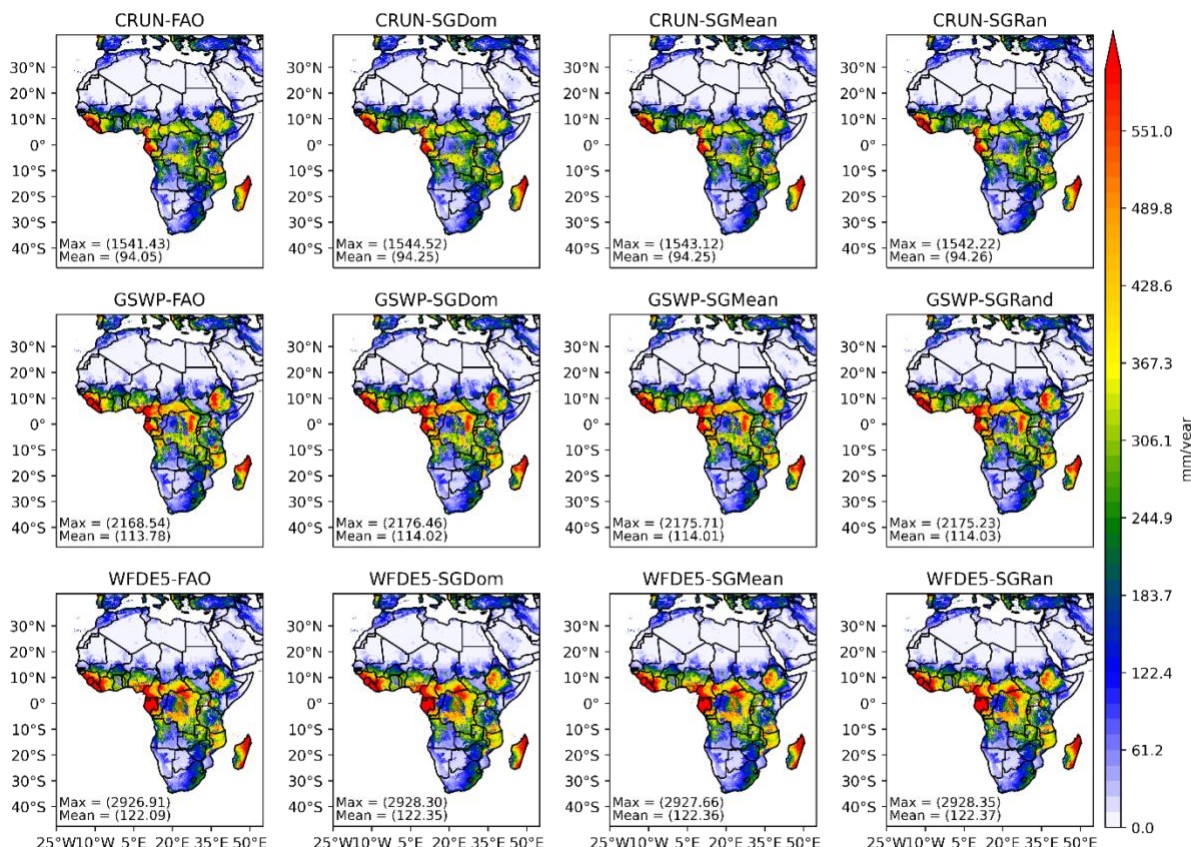

**Figure 5: Spatial distribution of simulated mean annual surface runoff over Africa. Upper row: CRUNCEP forced simulations with, from left to right, FAO, Dominant, Mean and Random upscaled soil texture map inputs. Middle row:**
**similar to row 1, but GSWP forced simulations. Bottom row: similar to row 1, but WFDE5 forced simulations.**

### 3.2.3 Subsurface runoff

Simulated subsurface runoff across the African continent (Figure 6) is in general low in most regions and across all simulation scenarios, typically below 250 mm/year. The estimation of subsurface runoff is more influenced by
soil texture variations and the upscaling of soil texture properties compared to ET and surface runoff simulations. The most substantial differences in simulated subsurface runoff among soil texture inputs are between the FAO soil map and the SoilGrids250m maps, while the disparities among the upscaled SoilGrids250m maps are smaller

especially with GSWP and CRUNCEP forcings. For CRUNCEP forcings, the difference between the maximum and minimum simulated subsurface runoff among the soil texture maps (averaged over Africa) is 11.3 mm/year, whereas it is 2.1 mm/year among the upscaled SoilGrids maps. For GSWP, these differences are 11.6 mm/year and 2.4 mm/year, respectively, while for WFDE5, they are 26.0 mm/year and 14.5 mm/year, respectively. Notably, for WFDE5 (with 1-hourly forcings), the differences in simulated subsurface runoff among the different upscaled SoilGrids maps are considerably larger than for the other forcings. The variations in maximum subsurface runoff values among soil texture maps are more pronounced than for the mean subsurface runoff, particularly for CRUNCEP and WFDE5, where the differences among upscaled SoilGrids maps are also substantial.

On the other hand, the spatially averaged subsurface runoff over Africa showed considerable variations among atmospheric forcings: 17-29 mm/year for CRUNCEP, between 36 and 48 mm/year for GSWP and 42-68 mm/year for WFDE5. Like surface runoff patterns, WFDE5 has the highest values, followed by GSWP, while CRUNCEP simulations yield the lowest subsurface runoff estimates. This discrepancy can be attributed to the higher average precipitation in WFDE5 over Africa (see Figure 2).

In summary, for subsurface runoff simulation, both variations in atmospheric forcings and soil texture, including different upscaling methods, play an important role.

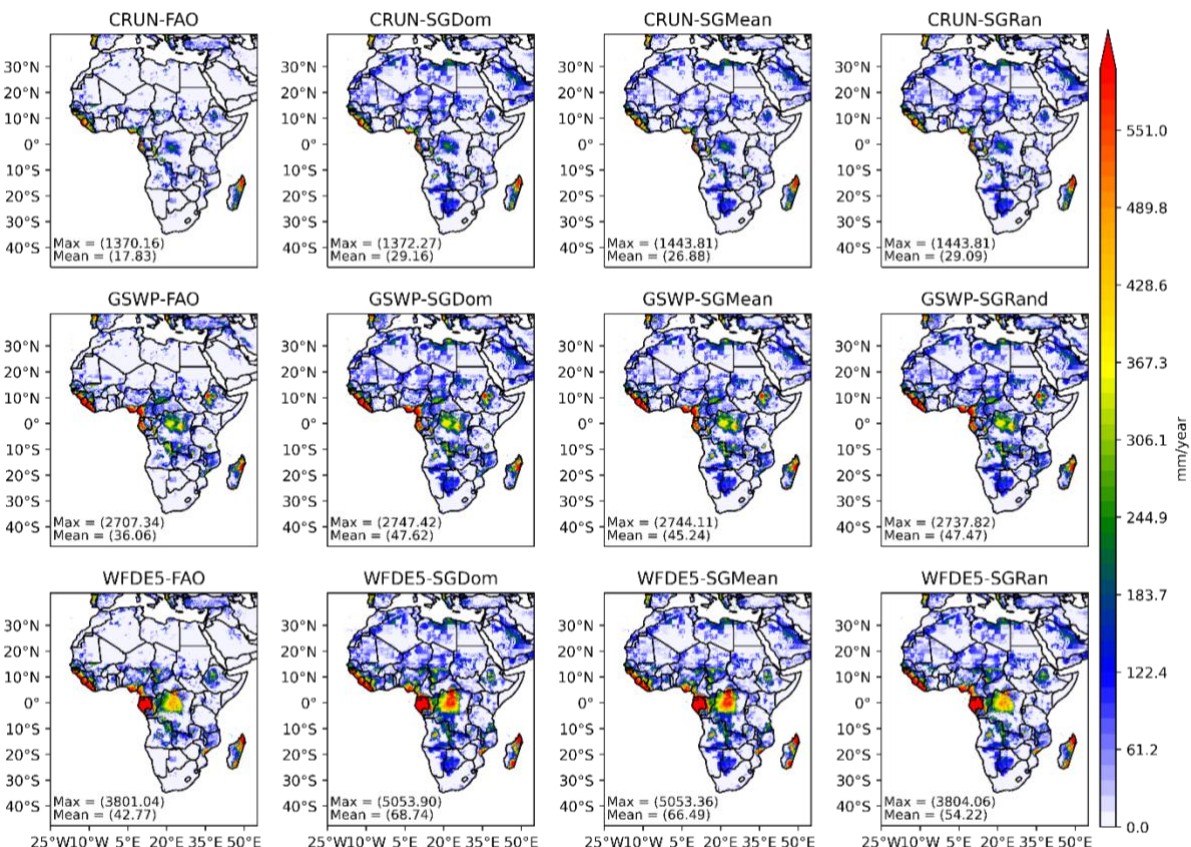

**Figure 6: Spatial distribution of simulated mean annual sub-surface runoff over Africa. Upper row: CRUNCEP forced simulations with, from left to right, FAO, Dominant, Mean and Random upscaled soil texture map inputs. Middle row: like row 1, but GSWP forced simulations. Bottom row: like row 1, but WFDE5 forced simulations.**

### 3.2.4 Soil moisture content

Soil moisture content estimates were obtained by calculating the weighted average of soil moisture content over the top 2 meters of the soil profile in CLM5. Mean annual maxima and averages have also been analysed for each season, as seasonal analysis of soil moisture content reflects seasonal changes in hydrological processes (Myeni et al., 2019), and allows a better understanding of the relationship between vegetation and water availability (Huber et al., 2011). Specifically, for the boreal summer season (JJA), the average simulated soil moisture content

across the African continent varies between 0.02 $cm^3/cm^3$ in the Sahara and 0.54 $cm^3/cm^3$ in both Equatorial Guinea and the coasts of Sierra Leone among the 12 simulations (Figure 7). The upscaled soil texture maps give all very similar continental averages of soil moisture content for the summer season. The source of the soil texture maps (FAO vs SoilGrids) resulted in some variation in the continental soil moisture content averages. A difference map showing the difference between the SGMean and the 3 other soil texture maps (FAO, SGDom and SGRand)

for the same season (Figure S2) also shows clearly that while there is a 0.0 $cm^3/cm^3$ continental mean difference among the upscaled SoilGrids maps, there is an maximum difference of 0.19 $cm^3/cm^3$, minimum difference of -0.19 $cm^3/cm^3$ and a mean continental difference of 0.01 $cm^3/cm^3$ between FAO and SGMean. This suggests that the source of a soil texture map could influence soil moisture content estimates by a land surface model more than the upscaling procedure of the soil texture information. The WFDE5 atmospheric forcings are associated with

more variation in simulated soil moisture content among the 4 soil texture maps than the other atmospheric forcings. The mean soil moisture content and the difference maps for other seasons can be found in Figures S3 – Figure S8.

Like for ET and surface runoff, varying the atmospheric forcing impacted continental maximum of soil moisture content more than variations in soil texture input. CRUNCEP forced simulations (6 hourly timesteps) gave lower

maximum soil moisture content values (0.46-0.47$cm^3/cm^3$) than GSWP (3 hourly timesteps; 0.51-0.53 $cm^3/cm^3$) and WFDE5 (hourly timesteps; 0.50-0.54 $cm^3/cm^3$) forced simulations. This difference is likely attributed to lower precipitation amounts in the CRUNCEP forced simulations, combined with slightly higher ET values in comparison to simulations with the other forcings.

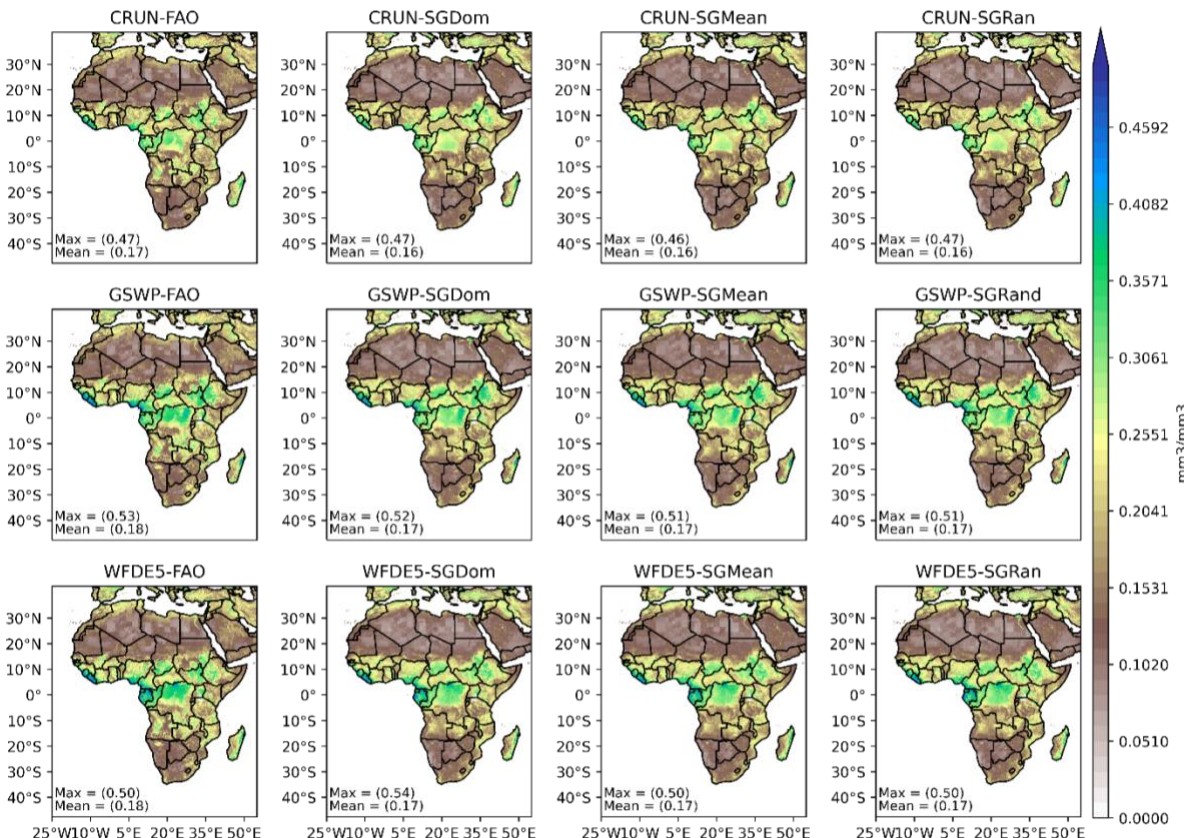

Figure 7: Spatial distribution of simulated soil moisture content in the JJA season over Africa. Top row: CRUNCEP forced simulations with FAO, Dominant, Mean and Random upscaled soil texture map inputs. Middle row: like top row, but GSWP forced simulations. Bottom row: like top row, but WFDE5 forced simulations.

## 3.3 Regional Results

We present results for two regions (Sahara and Central Africa) based on their moisture availability contrast.

### 3.3.1 Sahara region

The Sahara region is generally on average the most moisture deficient region in Africa. Rainfall over the region was highest in August 2013 (around 30mm/month) and was near 0mm/month for many other months, especially in the winter season (see Figure 8, row 1).

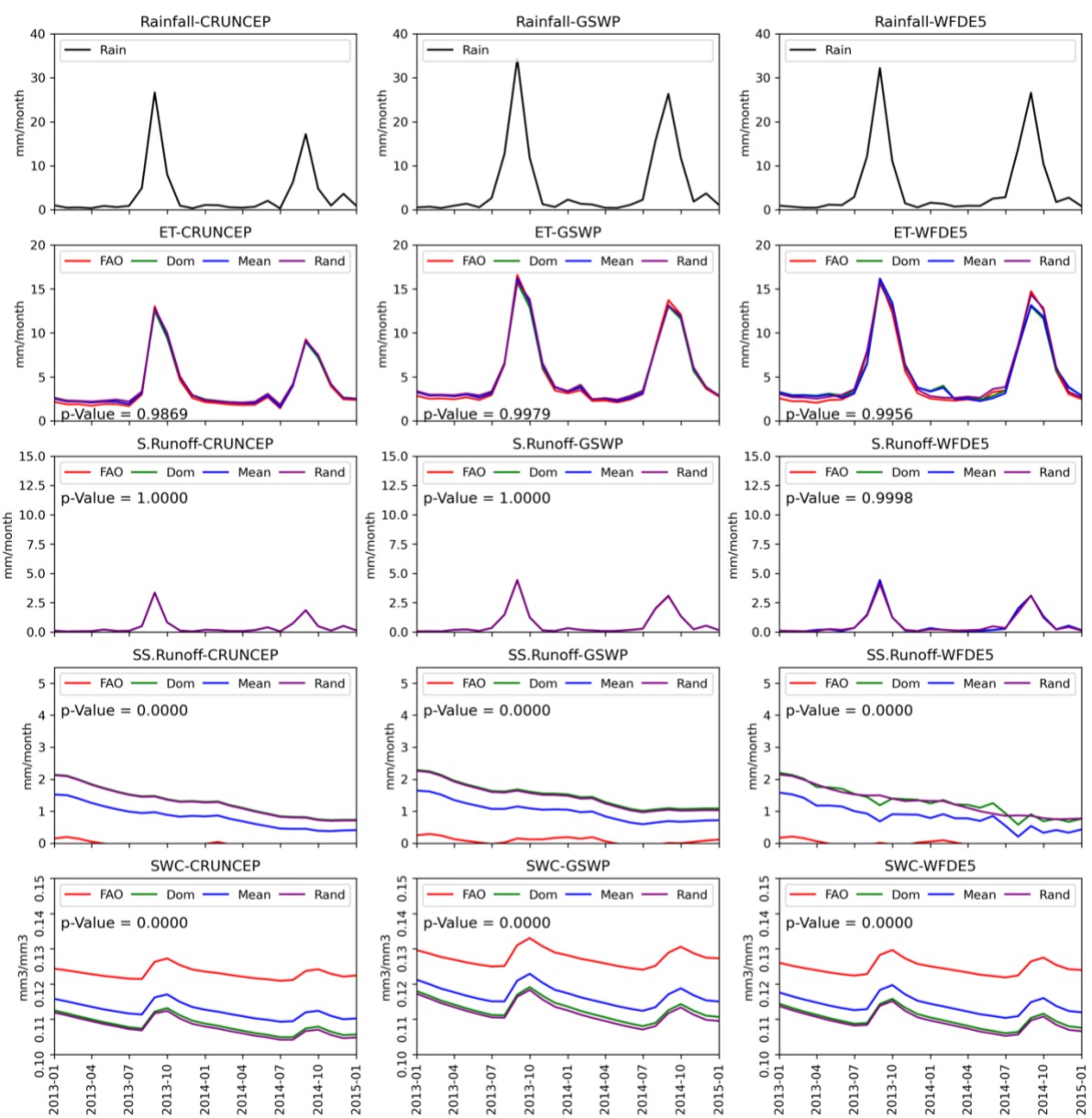


**Figure 8: Monthly regional mean of water balance components over the Sahara. The p-values indicate the statistical significance of the variations observed in the model outputs. Rows 1-5 show precipitation, actual ET, surface runoff, subsurface runoff, and soil moisture content respectively. Left, middle and right columns show the same variables for CRUNCEP, GSWP and WFDE5 atmospheric forcings respectively. The lines in the figures represent results for different soil textures as input. Red line: FAO, green line: SoilGrids-Dominant, blue line: SoilGrids-Mean and purple line: SoilGrids-Random.**

Simulated ET and surface runoff differed little among the different soil texture maps for the different atmospheric forcings. The average margin in actual ET among soil texture maps is only 0.4mm/month for both CRUNCEP and GSWP forcings, and 0.8mm/month for WFDE5 forcing. ET simulated by the CLM5 model varied more as function of the atmospheric forcing and can for a given soil texture map vary up to a few mm per month between different atmospheric forcings.

Simulated surface runoff exhibits similar patterns for all soil texture maps, with minimal surface runoff and slight increases during months with higher precipitation. The average monthly differences in surface runoff between the

different soil texture maps are smaller than 0.1mm/month. Subsurface runoff shows a decreasing trend, which is attributed to initially higher groundwater levels. While subsurface runoff is generally small in absolute terms, the different soil texture maps result in significantly varying relative amounts of subsurface runoff. Simulated average soil moisture content over the Sahara region is consistently low, with values around 0.12 $cm^3/cm^3$. These significantly different values, which are not extremely low despite very limited precipitation, could be attributed to the amount of loamy soil over the region (Figure S10) with higher residual soil moisture content than in sandy soils. Differences in simulated soil moisture content among the soil texture maps are primarily influenced by the variations in soil properties used in each map.

The different soil texture inputs to the WFDE5 forced simulations result in larger differences in simulated ET and surface runoff (though not significant according to ANOVA) compared to the other atmospheric forcings, for regions with low soil moisture content like the Mediterranean (Figure S12) and South-West Africa (Figure S16). The higher temporal resolution (1 hour) of the WFDE5 atmospheric forcing leads to varying surface runoff compared to forcings with lower temporal resolutions (3-hour or 6-hour).

Overall, these findings over the Sahara and other low moisture regions like the Mediterranean and South-West Africa highlight some influence of atmospheric forcing and its temporal resolution, soil texture maps variation, and their interactions on the simulation of ET, surface runoff, subsurface runoff, and soil moisture content across different regions of Africa.

**3.3.2 Central Africa**

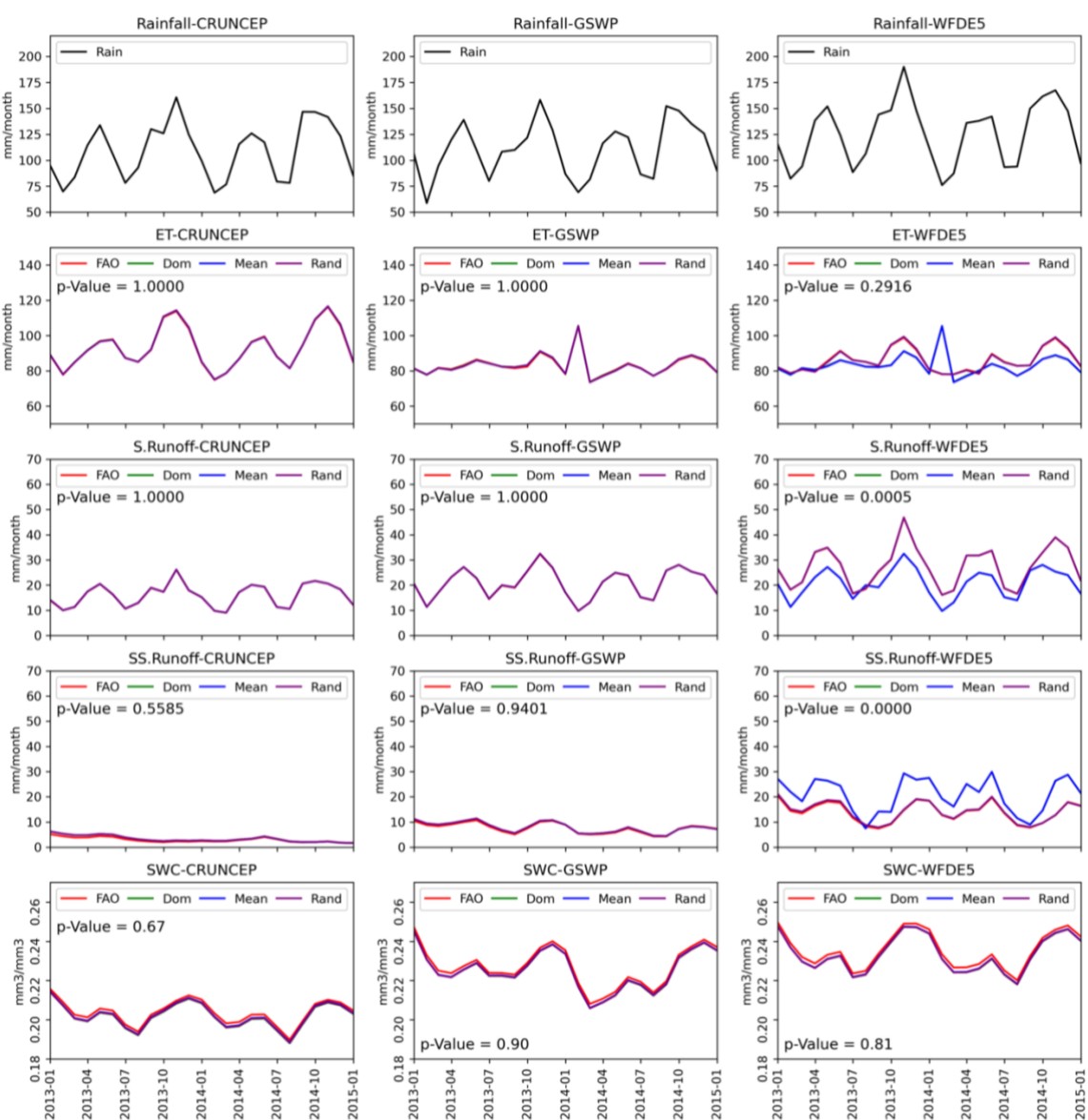

**Figure 9: Monthly regional mean of water balance components over the Central Africa. The p-values indicate the statistical significance of the variations observed in the model outputs. Left, middle and right columns show the same variables for CRUNCEP, GSWP and WFDE5 atmospheric forcings respectively. The lines in the figures represent results for different soil textures as input. Red line: FAO, green line: SoilGrids-Dominant, blue line: SoilGrids-Mean and purple line: SoilGrids-Random.**

Central Africa encompasses the Congo rainforest, the second-largest rainforest in the world, consisting of evergreen and semi-evergreen deciduous forests (Aloysius & Saiers, 2017) and stands out as one of the most moisture-rich regions in Africa, characterized by a regional mean rainfall ranging from 50 to 200mm/month. The proximity to the equator results in frequent rainfall events due to recurrent convective precipitation events. The dense vegetation in Central Africa contributes to high transpiration rates, which are supported by the substantial amounts of rainfall.

Once again, we observe that only the WFDE5 atmospheric forcings exhibits a variation (not significant) in ET values across different soil texture maps, as shown in Figure 9. On average (over the years 2013 and 2014), the

soil texture maps with the highest and lowest monthly averaged ET differ by 0.5mm/month for CRUNCEP and GSWP, but by 5.8mm/month for WFDE5. The monthly averaged surface runoff values for CRUNCEP and GSWP show little variation among different soil texture maps. However, for WFDE5, the SoilGrids map upscaled with random selection results on average in a significant 6.7mm/month higher surface runoff than the other soil texture maps. Regarding subsurface runoff, GSWP and CRUNCEP simulations exhibit, at most, a 0.4mm/month difference in average monthly subsurface runoff among different soil texture maps, whereas WFDE5 shows a significant difference of 7.0mm/month. The soil moisture content maps display near similar average values across all atmospheric forcings and soil texture maps with no significant differences.

Other moisture rich regions including West Africa (Figure S13), North-East Africa (Figure S14), Central-East Africa (Figure S15) and South-East Africa (Figure S17) also show that WFDE5 forced simulations resulted in clear differences which are mostly closer to significance than GSWP and CRUNCEP in simulated ET, surface runoff and subsurface runoff for the different soil texture inputs. On the other hand, soil moisture content did not show clear significant differences for the different soil texture maps in all regions.

### 3.4. Local Results

We now look at the results at the local scale (grid scale) to analyse further the impact of the variation of soil texture maps and atmospheric forcings on simulation outcomes. We selected one location for each of the eight climate regions: Cairo (Egypt, Mediterranean), Agadez (Niger, Sahara), Abuja (Nigeria, West Africa), Addis-Ababa (Ethiopia, North-East Africa), Salong (DR Congo, Central Africa), Daar-es-Salaam (Tanzania, Central-East Africa), Windhoek (Namibia, South-West Africa) and Maseru (Lesotho, South-East Africa). Two of the eight locations are discussed due to their contrasting moisture availability while other locations are available in the supplementary materials.

### 3.4.1 Agadez

Agadez, situated at 16.97°N and 7.98°E, experienced its highest precipitation of 125mm/month in August 2013 and received no rainfall during several winter months within our reference period. The grid cell in focus is also dominated by sandy and loamy soils according to all 4 soil texture maps. The results for Agadez indicate a close association between ET peaks and precipitation peaks, as ET in this region, the Sahara, is limited by water availability. Despite a five-month period without rainfall from September 2013 to January 2014, ET values in Agadez remained nonzero (1.5mm/month) between January 2014 and April 2014. This can be attributed to irrigation practices automatically applied to sustain irrigated crops when the soil moisture content falls below a critical threshold within CLM5.

The WFDE5 forced simulations for Agadez (Figure 10) show that different (upscaled) soil texture maps yield varying monthly ET, surface runoff, and subsurface runoff values. On average (over the years 2013 and 2014), the soil texture maps with the highest and lowest monthly averaged ET differ by 0.7mm/month for CRUNCEP, 0.9mm/month for GSWP, and 1.8mm/month for WFDE5 (though not significant according to ANOVA). This can be attributed to low rainfall. Model simulations driven by CRUNCEP or GSWP show no variation in surface runoff as function of the soil texture map, while slight but insignificant variations in surface runoff are found for WFDE5. A similar pattern is observed for subsurface runoff. Although the texture class for Dom and Mean is

605 Loamy Sand (LS) and for FAO and Rand Sandy Loam (SL) for the grid cell under concern (Table T17), statistically significant differences in soil water content are observed among the four soil texture maps. These differences arise because, although the soil texture classes are similar, the proportions of clay, sand, and silt vary among the four maps, resulting in different hydraulic conductivities.

Overall, the results for Agadez demonstrate the influence of soil texture map variation on ET, surface runoff and 610 subsurface runoff, with WFDE5 simulations exhibiting more pronounced variations compared to CRUNCEP and GSWP forcings. These findings underscore the importance of soil texture representation and temporal resolution of the atmospheric forcing in capturing the hydrological processes in Agadez and similar locations.

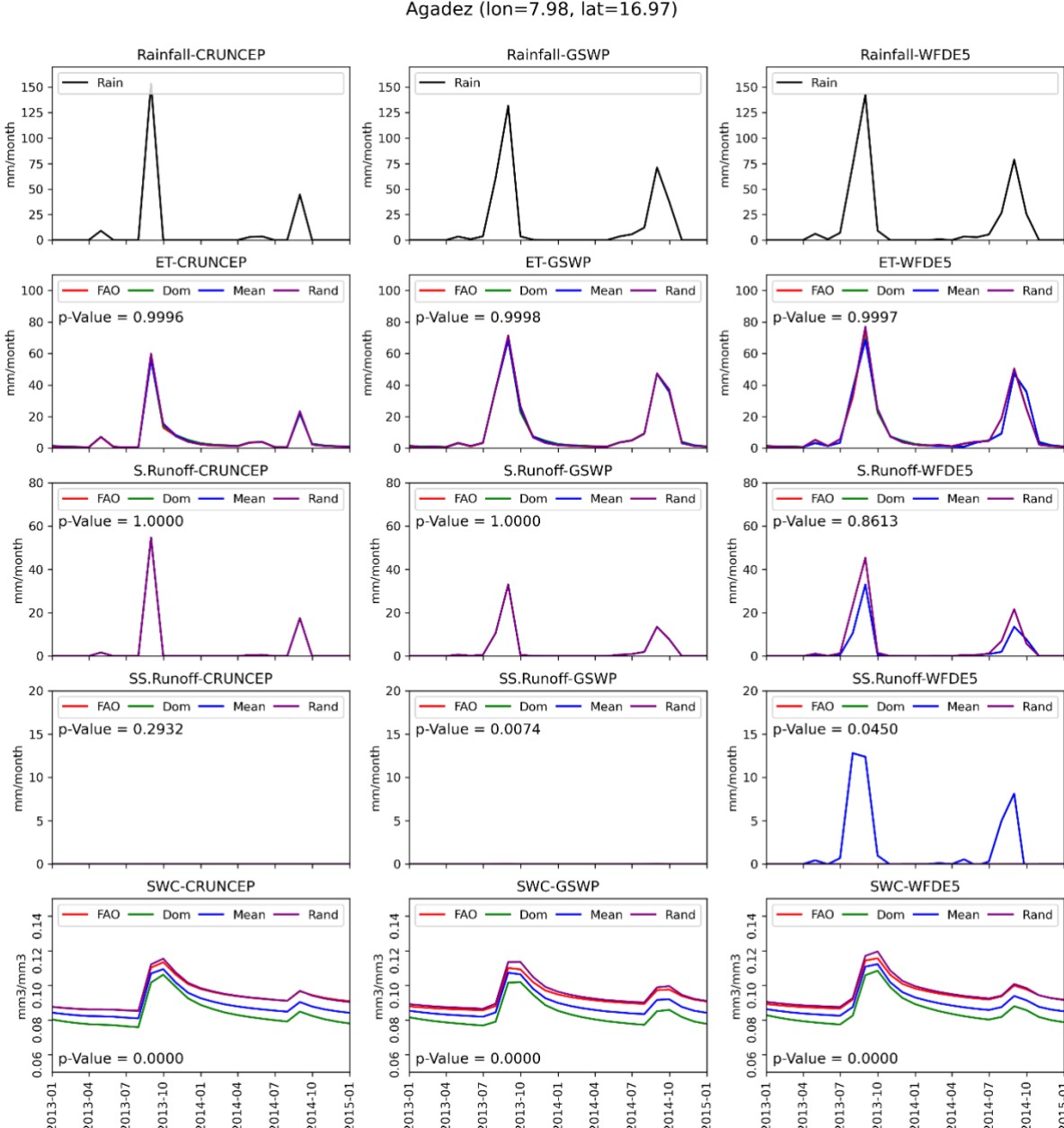

**Figure 10: Monthly local estimates of water balance components over Agadez. The p-values indicate the statistical 615 significance of the variations observed in the model outputs. Left, middle and right columns show the same variables for CRUNCEP, GSWP and WFDE5 atmospheric forcings respectively. The lines in the figures represent results for**

**different soil textures as input. Red line: FAO, green line: SoilGrids-Dominant, blue line: SoilGrids-Mean and purple line: SoilGrids-Random.**

### 3.4.2 Abuja

Abuja, situated in Nigeria at coordinates 9.07°N, 7.30°E, exhibits a distinct yearly precipitation cycle characterized by high rainfall during the summer months, with precipitation exceeding 200mm/month. Conversely, the winter season is dry, with months devoid of any rainfall (Figure 11). ET peaks in Abuja typically occur approximately one month or more after the peak of rainfall, as observed in 2014.

Simulated ET, surface runoff, and subsurface runoff in Abuja demonstrate variations across different soil texture maps, although not statistically significant, particularly noticeable with the high temporal resolution atmospheric forcings provided by WFDE5. In terms of ET, WFDE5 displays the highest mean margin differences among soil texture maps (10.4mm/month), followed by GSWP (1.4mm/month) and CRUNCEP (1.0mm/month). Regarding surface runoff, WFDE5 also yields the highest mean margin (7.5mm/month), while CRUNCEP and GSWP exhibit negligible differences (<0.2mm/month). Similarly, the ranking of differences in subsurface runoff follows the same pattern, with WFDE5 showing the largest disparities (7.7mm/month), followed by GSWP (0.4mm/month), and finally CRUNCEP (0.0mm/month). Notably, the FAO soil texture map consistently results in slightly higher soil moisture content (SWC) compared to the SoilGrids soil texture maps for all atmospheric forcings (as depicted in the lower row of Figure 8). However, these differences in SWC do not exceed 0.01 cm$^3$/cm$^3$ across all atmospheric forcings and are valued as insignificant according to ANOVA.

Similar patterns are observed for other locations, including Cairo (Figure S18), Addis-Ababa (Figure S19), Salong (Figure S20), Daar-es-Salaam (Figure S21), Windhoek (Figure S22), and Maseru (Figure S23). WFDE5 forced simulations exhibit larger variations in simulated ET, surface runoff, and subsurface runoff among different soil texture maps compared to the other atmospheric forcings. Simulated soil moisture content shows minimal variations among the different soil texture maps for a given atmospheric forcing.

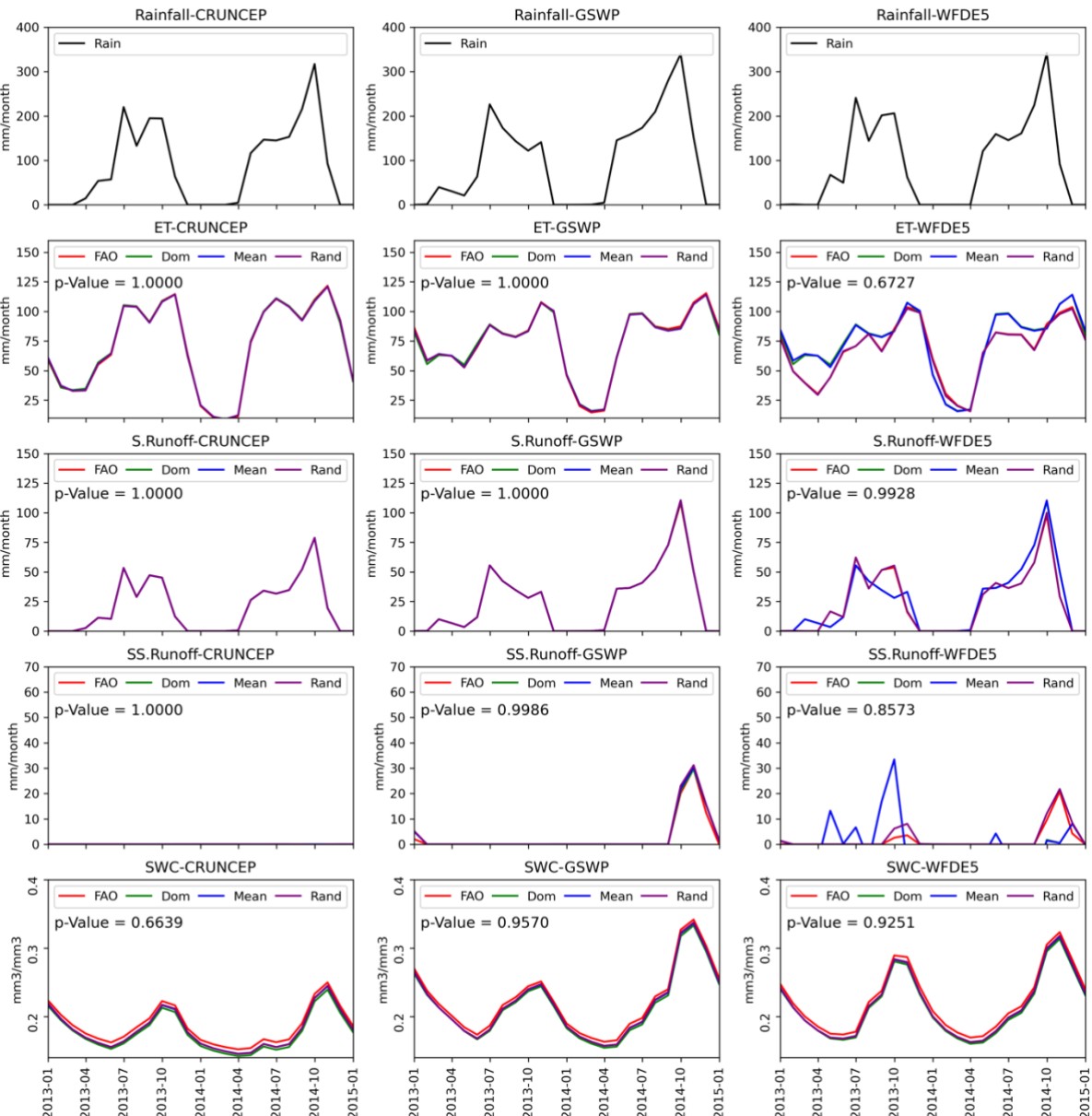

Abuja (lon=7.30, lat=9.07)

**Figure 11: Monthly local estimates of water balance components over Abuja. The p-values indicate the statistical significance of the variations observed in the model outputs. Left, middle and right colums show the same variables for CRUNCEP, GSWP and WFDE5 atmospheric forcings respectively. The lines in the figures represent results for different soil textures as input. Red line: FAO, green line: Soilgrids-Dominant, blue line: SoilGrids-Mean and purple line: SoilGrids-Random.**

### 3.4.3 Aggregation of WFDE5 to 3-hourly and 6-hourly

To further validate the role of the temporal resolution of atmospheric forcings, WFDE5 forcing was aggregated (from hourly data) to 3 hourly and 6 hourly so that it varied temporally only on a 3 hourly and 6 hourly basis, like GSWP and CRUNCEP. Simulations were performed to examine the impact of the new temporal resolution on ET, surface runoff, subsurface runoff and soil moisture content both regionally and locally. We compare CRUNCEP (6 hourly), 6H-WFDE5 (6 hourly), 3H-WFDE5 (3 hourly), GSWP (3 hourly) and WFDE5 (1 hourly).

Table 3: Mean margin of simulated variables among soil texture maps for CRUNCEP (6 hourly), GSWP (3 hourly), 6H-WFDE5 (6 hourly), 3H-WFDE5 (3 hourly) and WFDE5 (1 hourly)

| Variable | CRUNCEP | GSWP | 6H-WFDE5 | 3H-WFDE5 | WFDE5 |
|---|---|---|---|---|---|
| Evapotranspiration (mm/month) | 0.59 | 0.60 | 0.61 | 0.61 | 4.34 |
| Surface Runoff (mm/month) | 0.06 | 0.07 | 0.08 | 0.09 | 2.68 |
| Subsurface Runoff (mm/month) | 0.49 | 0.48 | 0.56 | 0.59 | 3.00 |
| Soil moisture content $cm^3/cm^3$ | 0.00 | 0.00 | 0.00 | 0.00 | 0.00 |

660    Table 3 shows the impact of varying soil texture map inputs on different water balance component for West Africa. Simulated variables show much less variation as function of soil texture map input for CRUNCEP, GSWP, 6-hourly aggregated WFDE5 forcings and 3-hourly aggregated WFDE5 forcings compared to 1-hourly WFDE5 forcings. Similar results are found for Abuja (Table 4), where CRUNCEP, GSWP, 6H-WFDE5 and 3H-WFDE5 forcings produce variations between 0.02 and 2.0 mm/month in ET, surface runoff and subsurface runoff as

665    function of the soil texture map, while WFDE5 produces variations between 7.43 and 9.93 mm/month among soil texture maps. Similar observation was also made for other regions (Tables S1, S2, S4, S5, S6, S7 and S8) and locations (grid cells) (Tables S9, S10, S11, S12, S13, S14, S15 and S16).

Table 4: Mean margin of simulated variables among soil texture maps for CRUNCEP (6 hourly), GSWP (3

670    hourly), 6H-WFDE5 (6 hourly), 3H-WFDE5 (3 hourly) and WFDE5 (1 hourly).

| Variable | CRUNCEP | GSWP | 6H-WFDE5 | 3H-WFDE5 | WFDE5 |
|---|---|---|---|---|---|
| Evapotranspiration (mm/month) | 0.91 | 1.33 | 1.70 | 2.00 | 9.93 |
| Surface Runoff (mm/month) | 0.03 | 0.15 | 0.19 | 0.21 | 7.43 |
| Subsurface Runoff (mm/month) | 0.02 | 0.41 | 0.59 | 0.63 | 7.70 |

| | | | | | |
|---|---|---|---|---|---|
| Soil moisture content cm$^3$/cm$^3$ | 0.01 | 0.01 | 0.01 | 0.01 | 0.01 |

### 3.5. Discussion

The simulation results over Africa suggest that the atmospheric forcings exert an important control on the ET estimates, while soil texture is important for simulating surface and subsurface runoff. The simulation results also suggest that the temporal resolution of atmospheric forcings influences the simulation outcomes, especially surface and subsurface runoff, and the interaction with soil texture seems to play a role here. These findings agree with the work of Zhang et al. (2023) on the role of soil texture and Beusekom et al. (2022) on the impact of temporal forcing aggregation on Land Surface model outputs.

The analysis of water budget components shows differences in simulated ET, surface runoff and subsurface runoff for the different upscaled soil texture maps in combination with WFDE5 forced simulations, but not in combination with other atmospheric forcings with more coarse temporal resolution. We observed for ET and surface runoff across all regions, that higher temporal resolution led to higher differences in ET and surface runoff between soil texture map outcomes with the largest differences for WFDE5 (hourly resolution), followed by GSWP (3 hourly resolution) and CRUNCEP (6 hourly resolution). For subsurface runoff, higher temporal resolution did not result in such a systematic pattern in moisture-rich regions with rainfall above 200mm/month. However, in moisture deficient regions higher temporal resolution of atmospheric forcing is associated with more variation in subsurface runoff for different soil texture maps. The temporal resolution of the atmospheric forcings did not result in different soil moisture content results for each soil texture map, but in all regions it was observed that the FAO soil texture map resulted in different soil moisture content than the Soilgrids250m soil texture maps partially confirming the findings of Tafasca et al. (2019) which showed that soil mapping had a stronger influence on soil moisture content compared to fluxes.

#### 3.5.1 The role of temporal resolution in rainfall intensity representation

We investigated whether the higher temporal resolution of simulations influenced the rainfall partitioning into surface runoff and infiltration. The absolute monthly (Figures S26 and S27) and annual (Figure S9) precipitation amounts over the continent vary only slightly among CRUNCEP, GSWP and WFDE5. The spatial averages for annual precipitation are 608mm/year, 638mm/year and 666mm/year for CRUNCEP, GSWP and WFDE5 respectively. These differences in rainfall amount do not explain why only for WFDE5, soil texture variations result in larger runoff and evapotranspiration variations. We analysed also the number of precipitation events with a rainfall intensity above 3mm/hour for each of the three atmospheric forcings and eight selected locations. We found that WFDE5 had a much higher number of precipitation events with rainfall intensity greater than 3mm/hour than both CRUNCEP and GSWP at all 8 locations (see Table T17) indicating a better representation of rainfall intensity. GSWP and CRUNCEP had more rainfall events with much lower intensities. This indicates that rainfall intensity representation and its impact on the partitioning between infiltration and surface runoff in the land surface model is a likely reason for the higher sensitivity of model outcomes towards soil texture input in WFDE5 forced simulations than GSWP and CRUNCEP forced simulations.

**3.5.2 The role of soil texture in water balance components**

Rainfall intensity has a stronger influence on surface runoff generation than rainfall amount (e.g., Jungerius & ten Harkel, 1994; Yao et al., 2021) and surface runoff is on the other hand also strongly influenced by the hydraulic conductivity with lower conductivity supporting higher surface runoff (Suryatmojo & Kosugi, 2021; Ow & Chow, 2021; Chandler et al., 2018). Therefore, for WFDE5 forcings there are potentially more situations with surface runoff, such that the role of different soil properties can come into play. We analysed this for all 8 locations (Figure S53) by calculating the standard deviation of the fraction of precipitation turned into surface runoff among the 4 soil texture maps, for each atmospheric forcing. For the WFDE5 atmospheric forcings, this standard deviation varies between 1.2% of rainfall for Daar es Salaam and 10.1% of rainfall in Addis-Ababa while the standard deviations are less than 0.4% for both CRUNCEP (6 hourly) and GSWP (3 hourly) atmospheric forcings, for all locations. This identified impact of surface runoff agrees with Mizuochi et al. (2021) for the ORCHIDEE model and Fersch et al. (2020) for the WRF-Hydro model. This shows that the soil texture information has a control on the partitioning of fluxes for higher temporal resolution atmospheric forcings (Shuai et al., 2022). Since surface runoff and infiltration are sensitive to rainfall intensity (Mertens et al., 2002) and soil texture determines saturated hydraulic conductivity and therefore the timing of runoff (Hammond et al., 2019), surface runoff and subsurface runoff vary as a function of soil texture inputs in the WFDE5 simulations (mainly at the local and regional scales).

**3.5.3 Implications for land surface modelling and community impact**

This work demonstrates the critical role that high-resolution soil texture information and higher temporal resolution forcing datasets play in simulating water balance components. It highlights the need to use higher resolution soil texture information in land surface model simulations to improve the capturing of grid and sub-grid scale land surface heterogeneity. It is also necessary to provide better pedotransfer functions which link soil texture and soil hydraulic parameters which ultimately control infiltration. Higher temporal resolution of atmospheric forcing (hourly) in this work has also captured water balance dynamics differently from coarse temporal resolution atmospheric forcing which indicate a need for the community to further strengthen research to improve temporal resolution of atmospheric forcings especially over Africa. There have been advances in improving spatial resolution of atmospheric forcings (e.g., Funk et al., 2015) but this work serves as an indicator that higher temporal resolution atmospheric forcings are also needed. The works of Hersbach et al. (2020) and Cucchi et al. (2020) must be complemented in producing higher temporal resolution of atmospheric forcings. This advancement can eliminate the need for temporal disaggregation of precipitation as done in this work. This work showed that soil texture information is important in combination with high temporal resolution of atmospheric forcings as it impacts the division of rainfall into surface runoff and infiltration. Ultimately, land surface models also need to be better tuned to correctly reproduce this division, in the context of the higher temporal resolution of atmospheric input data and higher spatial resolution of information on soil hydraulic properties.

**4. Conclusion**

1.  Community Land Model version 5 (CLM5) model runs over the African continent were performed at a high spatial resolution of approximately 3km, with four different soil texture maps and three different atmospheric forcings. The four different soil texture inputs included the FAO soil map and three

differently upscaled SoilGrids250 maps. The three different atmospheric forcings were CRUNCEPv7, GSWP3 and WFDE5. The most important findings were: Average evapotranspiration and surface runoff simulated by CLM5 over the African continent show a limited sensitivity to variations in the soil texture input. The source of soil texture information (FAO versus SoilGrids) results only in minor variations in the continental average ET or surface runoff (0.3% variations around mean), and the impact of different upscaling approaches of soil texture information is even smaller. This sensitivity to soil texture input is much smaller than the sensitivity to the different atmospheric forcings (3% variations for mean ET and 26% for surface runoff). Average subsurface runoff and average soil moisture at the continental scale are both as sensitive to variations in atmospheric forcings as to variations in soil texture information.

2.  Although average surface runoff at the continental scale shows a limited sensitivity to soil texture input, at the regional and, especially, the local scale this sensitivity is much higher, but mainly in combination with the higher temporal resolution of WFDE5 forcings (hourly). The higher temporal resolution of WFDE5 forcings (hourly) than the other atmospheric forcings resulted not only in larger variations in simulated surface runoff, but also ET and subsurface runoff for the different soil texture maps. This points to the fact that the impact of soil texture becomes more important in combination with higher temporal resolution of atmospheric forcings. We explain this with the impact of the temporal resolution of atmospheric forcings on the rainfall intensity and the partitioning of rainfall into surface runoff, which is also determined by the hydraulic conductivity of the soil. This, in turn, affects also the amount of water available for evapotranspiration and drainage.

This study therefore recommends further advances in the provision of both higher temporal resolution climate datasets and higher spatial resolution soil information over Africa. With higher spatial resolution soil information, sub-grid scale land surface heterogeneity will be handled with more accuracy. Also, higher temporal resolution climate datasets at less than 1-hour timesteps will not only eliminate the need for temporal disaggregation in land surface model applications but ensure that more accurate atmospheric variables are supplied to the land surface model at each time step.

This study also highlights specific implications for the simulation of surface runoff by land surface models. Higher spatial resolution of soil texture data, or soil hydraulic properties, at finer spatial scales allow potentially for a better modelling of surface runoff and subsequently other water balance components at each grid cell. In addition, higher temporal resolution atmospheric forcing captures high-intensity rainfall events that can produce more surface runoff in a short period of time, especially on soils with low hydraulic conductivity, leading to a more accurate estimate of surface runoff at each affected grid cell.

It is assumed in this work that model shortcomings (for example related to the representation of yearly vegetation cycles and the representation of different crop types) do not affect substantially the differences in the simulation results. Furthermore, the release of CLM5 used in this work assumes 16 plant functional types for the African continent by default which does not represent all vegetation types. Also, irrigation is hardcoded into the surface datasets. Future work should reduce these limitations.

## Data Availability

This study only uses publicly available data sources, and they were cited wherever applicable. SoilGrids dataset
was obtained from https://files.isric.org/soilgrids/former/2017-03-10/data/ last opened: 18.06.2024, WFDE5 was
obtained from https://cds.climate.copernicus.eu/cdsapp#!/dataset/derived-near-surface-meteorological-
variables?tab=overview last opened 18.06.2024. GLDAS-2.1 was obtained from
https://disc.gsfc.nasa.gov/datasets/GLDAS_NOAH025_M_2.1/summary?keywords=GLDAS last
opened:07.07.2024 while both CRUNCEP and GSWP datasets were downloaded from NCAR's publicly
accessible repository: https://svn-ccsm-inputdata.cgd.ucar.edu/trunk/inputdata/atm/datm7/ last opened
18.06.2024. Summary tables indicating average margin for simulated variables among soil texture maps for
GSWP, WFDE5 (3 hourly) and WFDE5 (hourly) as calculated are included in the supplement. The coordinates
for partitioning Africa into 8 regions as done in this study were also cited where applicable.

## Author Contribution

Bamidele Oloruntoba: Data curation, Methodology, Formal analysis, Investigation, Software, Visualization,
Writing – original draft preparation. Stefan Kollet: Conceptualization, Methodology, Writing – review &
editing. Carsten Montzka: Conceptualization, Methodology, Writing – review & editing. Harry Vereecken:
Conceptualization, Methodology, Writing – review & editing. Harrie-Jan Hendricks Franssen:
Conceptualization, Methodology, Supervision, Writing – review & editing.

## Competing Interests

At least one of the (co-)authors is a member of the editorial board of Hydrology and Earth System Sciences.

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
