# Peer review of "High Resolution Land Surface Modelling over Africa: the role of uncertain soil properties in combination with forcing temporal resolution."

_EGUsphere, 2023_

## Referee Comment (RC2)

**Review Comments (egusphere-2023-3132)**

Oloruntoba et al. analyzed how various soil data and forcings impact CLM5 water-related variable simulations across Africa. This work could be valuable in guiding future simulations. However, the paper is not well-written. Below, I list some of my comments for improvement.

1. The title is confusing. What do you mean by "temporal model resolution"? It needs to be modified. It does not reflect the important points mentioned in your abstract about different soil textures and forcing sources, which are crucial to your paper.

2. The abstract is not well organized. From your summary of your experiments (4 soil textures and 3 forcings), readers might expect results on how soil impacts the simulations and how different forcings affect the simulations. However, the abstract does not mention how different forcings affect your simulations, i.e., the IV point in your conclusion section. Thus, I think the abstract is not well organized.

3. The introduction should be reorganized or rewritten. Many paragraphs in the introduction belong in the method section. For example, paragraphs 4 through 6, about specific soil data, forcing, and experiment design, respectively, should be mentioned in the methodology section. Therefore, the introduction should be more thoughtful and include more logical content with citations. As you have written the method in the introduction section, it makes some information in the method section somewhat fragmented; for example, L275 mentions "The four (4) soil texture maps", which originally comes from L81 of the introduction.

4. In L265, please clarify if one year of spin-up is enough. Running the entire domain for a long time may be challenging, but you need to check if one year is sufficient. I suggest picking several typical grid cells with typical plant functional types and/or soil textures, to test if the one-year spin-up is enough for your analyzed variables (ET, runoff) to become stable enough to show the difference between your 12 different experiments. That is to say, can your conclusion made here represent results if run the simulations for 30 years?

5. For the different soil texture upscale data, is it common in our community to "randomly select a single SoilGrid cell"? If so, please provide some references; if not, why do you want to test it here if few uses it in the community?

6. L274. Is the metric termed "average margin" developed by you? If not, you need to add proper citations.

7. Figure 2 could be moved to the supplementary because it is not part of your main story but an explanation for your main story.

8. It is better to use some quantitative metrics to quantify the differences in model simulations between different experiments, instead of a qualitative way, to distinguish which factors (e.g., different soil, different forcing) are the most important.

9.  The analysis would be better if further compared with benchmark datasets.

10. L52. Correct the typo of the degree symbol from "0.5o", and check for other similar errors throughout the document.

11. In the Data Availability section, it would be better to include where (URL or platform) you specifically obtained the data, e.g., SoilGrids, different forcings, etc.

12. The discussion section includes numerous analyses which, although sufficient, feel oddly placed. If there are so many analyses based on new tables, why not include them in the results section (e.g., Tables and new figures)? Additionally, the discussion does not provide a broad scope that shows how your study could further connect with or guide future research.

---

## Author Comment (AC1)

Reviewer's comment in bold, author's response in non-bold blue text.

This study uses the CLM5 model over Africa with different atmospheric forcings and soil texture inputs and explores their role in estimates of ET, runoff and soil moisture. Overall, there is a larger effect from the forcing dataset than from the soil texture inputs.

**General comments**

In my view, the paper in its current form requires major revisions before it can be considered for publication. The authors should specifically address the following limitations:

- **Since the study period is relatively short (two years): how can we know the results are generalizable to other years? Would it be an option to extend the study period?**

  This work investigates the interaction between uncertain soil properties and the temporal resolution of atmospheric forcings at the continental, regional and local scale with a focus on the rainfall intensity representation in the land surface model input. It therefore suggests that as long as the CLM5 model does not handle precipitation differently, extending the study period may not change the current outcome.

- **The study misses an evaluation with reference data of ET, soil moisture, and runoff to better appreciate the impact of the work: which forcing and soil texture input combination do the authors recommend the community to use in future work? Section 3.3 (local results) could be dropped in favour of this.**

  Performing an evaluation with measurement data is beyond the scope of this paper as stated in lines 60-61. The aim of the paper is not to select the best model settings over Africa or extensively evaluate the model performance, but to investigate the impact of certain model inputs (different soil input data versus different atmospheric forcings and role of temporal resolution) in a more generic sense. However, we will present a comparison of our ET, runoff and soil moisture estimates with widely available gridded datasets.

- **Often small impacts are extensively discussed, but it's not clear whether they are significant or not. Statistical tests should be performed to verify this.**

  We will reformulate the text to take this comment into account and provide information on statistical significance if possible.

- **I have methodological concerns about using the "random selection" upscaling technique for a single grid cell and then comparing the results with**

**other upscaling techniques. Such results are not reproducible, so how should they be interpreted?**

Random results are reproducible using "seeding" in programming. Seeding is used to initialize the random number generator (RNG) such that it produces a reproducible sequence of numbers. We specified seeds before performing upscaling using the Random method. If we therefore perform upscaling using the Random method multiple times, we will have the same outcome.

- **The Methodology is too detailed in terms of equations that are used by the LSM. These equations are only useful if they are referenced to in the text to explain or discuss the results, which is currently not the case.**

  - We will link the discussion in the paper better to the method section and might also move some of the material presented in Methodology to an appendix,

- **The Discussion is too descriptive and fails to convincingly explain some of the results.**

  We will deepen the discussion and discuss the results more by making more reference to equations in the revised version.

**Specific comments**

1. **The title hints an impact of "temporal model resolution", but this should be the resolution of the atmospheric forcing dataset (input for the model). Temporal model resolution is always 30 minutes, and its impact was not examined.**

   Yes, Land Surface Model temporal resolution is usually 30 minutes. We will revise the title in the revised manuscript version for consistency. We are considering "High Resolution Land Surface Modelling over Africa: the role of uncertain soil properties in combination with forcing temporal resolution" in the revised version.

2. **L62-69: Part of this paragraph should go to the Methodology, for example, how the dataset was upscaled to the model resolution.**

   Lines 67-69 where upscaling is described will be moved to methodology in the revised version.

3. **L70: many new acronyms which should be written in full the first time they are introduced.**

   CRUNCEP (Climatic Research Unit (CRU) and National Centres for Environmental Prediction (NCEP)), GSWP (Global Soil Wetness Project) and WFDE5 (Watch Forcing Data methodology on ERA 5) will be written in full in the revised version.

4.  **L83: "the novelty of the work lies in ..."; this is more or less a repetition of line 58.**

    We will merge both sentences into lines 83-85 in the revised version.

5.  **L88: the Introduction is finished with raising two main research questions on which the authors want to find the answer. I expect to explicitly find an answer to question 1 and question 2 somewhere in the Discussion or (preferably) the Conclusions.**

    Answers to the research questions as shown by the work will be explicitly stated in the conclusions of the revised version of the manuscript.

6.  **L94: Add a citation to CLM5.0. Additionally, be consistent in the use of CLM5.0 and CLM5: both are currently found throughout the text.**

    CLM5.0 (Community Land Model version 5.0) will be written in full in L94 and thereafter referred to as CLM5. All other uses of "CLM5.0" within the text will be changed to CLM5 in the revised version.

7.  **L101: "CLM5.0 therefore has features of great interest for land surface modelling over Africa at a high resolution". Why specifically Africa?**

    We did not want to place the focus on Africa, but on the high spatial resolution. The sentence will be changed to clarify this.

8.  **L106: Acronym for CLM5 should be written out in full the first time it is used (in the Introduction).**

    This has been addressed in our response to comment 6.

9.  **L107: Change "The total porosity is given by" to "The total porosity for level i is given by".**

    This will be changed as you suggest in the revised version.

10. **L117: Citation for the Brooks and Corey model? What is the physical interpretation of "the exponent B"?**

    The B in the Brooks and Corey model represents the pore size distribution. The text "which represents the pore size distribution" will be added to line 117 in the revised version.

11. **Overall, the introduction has too many equations which are not referenced anywhere in the text. So it is not clear what their added value is. Either they should be used in the Discussion to support in explaining the observed results, or they should be replaced by a citation of the CLM5 paper in which they can probably also be found?**

We will link the discussion in the paper better to the method section and might also move some of the material presented in Methodology to an appendix.

12. **L162: Add that the time step Δt is expressed in seconds.**

The units for Δt (seconds) will be added to L162 in the revised edition.

13. **L167: What is the native resolution of IGBP-DIS? Is it 3km? If not, why is only SoilGrids250m upscaled?**

The IGBP-DIS soil texture information is 8km resolution. This was internally downscaled by CLM5 using the nearest neighbour algorithm.

14. **L170: "The 10 upper CLM5 soil layers": how many are there in total? This type of information on the model should be in section 2.1, rather than the equations which are currently there.**

There are 20 active soil layers in CLM5 (8.6m deep). In this work we are concerned with the top 2m. L170 will then be changed to "The soil texture dataset provides information for the top 10 CLM5 soil layers: at 0.0175, 0.0451, 0.0906, 0.1656, 0.2892, 0.493, 0.829, 1.3829, 2.2962 and 3.4332 meter depth." From "The soil texture dataset provides information for the 10 upper CLM5 soil layers: at 0.0175, 0.0451, 0.0906, 0.1656, 0.2892, 0.493, 0.829, 1.3829, 2.2962 and 3.4332 meter depth."

15. **L181-182: Too detailed and not relevant for this study.**

We will delete this sentence in the revised version of the manuscript.

16. **L200: Is method (iii) the novelty of this paper? If yes, it should be mentioned explicitly and motivated (why is a third new method necessary if two other already exist). If no, a citation to another study using this approach is missing.**

Yes, the "Random upscaling" method is a novelty of this paper as it is yet to be documented. While the Dominant upscaling method introduces bias towards the prevalent soil types and the Averaging method presents a conservative estimate that misses extremes, the random upscaling method, creates an opportunity for outliers to be the effective soil parameter. This detail will be included in the revised version.

17. **Table 1: the longitudinal extent is not relevant for this study: all three are global and thus cover the study domain.**

The longitudinal extent will be removed in the revised version.

18. **L256: "A spatially varying soil thickness dataset": which one?**

The spatially varying Soil Thickness dataset is new in CLM5. It replaces an assumed soil thickness of 2m present in earlier versions of the CLM. The reference to the dataset has been provided in L257.

19. **L270: "the hourly WFDE5 forcings were also aggregated to 3 hours and used in new simulations". Perhaps this should go to section 2.4. Why not 6h, so all three forcings are comparable?**

Aggregation of WFDE5 into 3 hourly timesteps is an additional model setup performed after initial results already confirmed the effects of the higher temporal resolution of the WFDE5 data. We reasoned that making WFDE5 similar to GSWP (3 hourly) was sufficient to validate our claim.

20. **L276: "A simulated variable for a certain atmospheric forcings- soil texture map combination at a given time step is denoted by M1(t), M2(t), M3(t) and M4(t)." Don't you have 12 different of such combinations? Or do you mean that you have 4 such combinations for each atmospheric forcing? Please be more specific in the definition. Also, define t as the time step.**

Yes, we have a combination of the 4 soil texture map outputs for each atmospheric forcing. The text on L276 stating "A simulated variable for a certain atmospheric forcings- soil texture map combination at a given time step is denoted by M1(t), M2(t), M3(t) and M4(t)." will be changed to "The average margin for a simulated variable for a certain atmospheric forcing/soil texture map combination at a given time step is denoted by M1(t), M2(t), M3(t) and M4(t)."

In L283, "where T represents the total number of time steps in the time series." will be changed to "where T represents the total number of time steps in the time series and t denotes time step."

21. **L290: ET is evapotranspiration, not evaporation.**

L290 will be changed to Evapotranspiration in the revised version.

22. **L298: "marginally" implies that you performed a statistical test. One should actually be performed to assess the significance of the differences in mean/max ET (and other variables), depending on the texture, forcing and their interaction.**

The word "marginally" used in L298 will be changed to "slightly" in the revised version.

23. **L330-332: again, the statements made here require hypothesis testing.**

The word "significantly" in the sentence will be changed to "substantially" in the revised version. We will provide information on statistical significance if possible.

24. **L335: WFDE5 are not hourly because they have been aggregated to 3h (L270)?**

At this stage, WFDE5 is still hourly. We will clarify this in the text.

25. **L337: "The increased surface runoff in the WFDE5 forced simulations reduces the availability of water for ET processes"; can this claim be made with certainty if it also rains more, so more water is coming in (cfr. Fig. 2)?**

   The statement "The increased surface runoff in the WFDE5 forced simulations reduces the availability of water for ET processes" will be changed to "The increased surface runoff in the WFDE5 forced simulations reduces the availability of water for ET processes especially during sheet flow."

26. **L339: "A higher temporal resolution of the atmospheric forcings as for WFDE5 will result in higher peaks of precipitation intensity, whereas a coarser resolution of 6 hours like for CRUNCEP will average out intensive precipitation over longer time periods with less high peaks in precipitation intensity"; but wasn't this effect to be mitigated by aggregating the hourly dataset (L270)?**

   This effect was mitigated in the validation set-up where WFDE5 was aggregated to 3 hourly. But in the initial run, which first indicated the finding, the statement holds. We will clarify the text accordingly.

27. **L353: "significantly influenced": be careful using this word.**

   The statement "The estimation of subsurface runoff is more significantly influenced by soil texture variations and the upscaling of soil texture properties compared to ET and surface runoff simulations." Will be changed to "The estimation of subsurface runoff is more substantially influenced by soil texture variations and the upscaling of soil texture properties compared to ET and surface runoff simulations." In the revised version.

28. **L355: "The most substantial differences in simulated subsurface runoff are observed between the FAO soil map the SoilGrids250m maps"; actually the forcings still result in the largest differences.**

   We will change this sentence to "The most substantial differences in simulated subsurface runoff among the soil texture maps are observed between the FAO soil map and the SoilGrids250m maps, while the disparities among the upscaled SoilGrids250m maps are smaller."

29. **L364: Fig. 4 should be referenced to at the start of the section.**

   We will change this as suggested by the reviewer.

30. **L371: It's confusing to talk about ET in the section on runoff, save this for the Discussion.**

   We will remove this sentence.

31. **L382: "weighted average": how are the weights defined? Perhaps this is something to be described in the Methodology.**

The weights are defined according to the thickness of each soil layer. In L267, the sentence "For continental scale results, yearly sum of evapotranspiration (ET), surface runoff, and subsurface runoff were computed as well as seasonal mean soil moisture contents." will be changed to "For continental scale results, yearly sum of evapotranspiration (ET), surface runoff, and subsurface runoff were computed as well as the seasonal mean of the weighted average of the top 2 meters soil moisture content. The weights were defined according to the thickness of each soil layer in CLM5"

32. **L394: This statement doesn't hold for the mean: similar effect when varying the forcing and the texture.**

L394 will be changed to "Like for ET and surface runoff, varying the atmospheric forcing impacted continental maximum of soil moisture more than variations in soil texture input."

33. **Figure 5:**

   o **Would it make sense to also look at min soil moisture, next to mean and max?**

   Minimum soil moisture will be displayed in the revised version.

   o **Units of the colorbar are mm³/mm³, while elsewhere in the text cm³/cm³ is used. Be consistent. Also, consider using more sensible intervals for the colorbar ticks.**

   Colorbar units for soil moisture will be harmonised and intervals made more relatable in the revised version.

   o **Perhaps one could keep the first column as is, and the other columns could show the difference w.r.t. the first column? Now all 12 maps look identical and it's hard to see where the differences are.**

   This will be done for soil moisture as a supplementary material and referred to within the text in the revised version.

34. **Section 3.2: Consider moving the definitions of the regions to the Methodology.**

The definition of regions will be moved to methodology in the revised version.

35. **L410: Motivate the choice for Sahara and Central Africa (dry and wet region?).**

Yes. The choice of the Sahara and Central Africa are based on their moisture availability. The sentence "Two regions are discussed while more information for

other regions is available as supplementary material." In L410 will be changed to "Two regions are discussed while more information for other regions is available as supplementary material. These two regions are shown here while they have strongly contrasting moisture availability."

36. **L417: This statement is true on average (it may well exceed this in certain pixels).**

   The statement "The Sahara region is the most moisture deficient region in Africa…" in L417 will be changed to "The Sahara region is generally the most moisture deficient region in Africa…" in the revised version.

37. **Figure 7:**

   o **"Soil water content", "soil moisture content", "soil moisture" are all used interchangeably. Be consistent.**

   We will adhere to using soil moisture content in the revised version.

   o **Dominant and Random are very similar, as can be expected. The first one has already been used in literature. Please motivate (in the Methodology) the advantage of Random to justify it as a new technique (e.g., are there computational gains?). Also: which one is best? Why was no evaluation performed? There are many global grid-based ET products available that can serve as reference.**

   While the Dominant upscaling method introduces bias towards the prevalent soil types, the random upscaling method, creates an equal opportunity for all grid cells, especially for outliers to be the effective soil property. This detail will be included in the revised version.

38. **L430: Why is the long name for ET suddenly introduced here?**

   This will be changed in the revised version.

39. **L434: "Subsurface runoff shows a decreasing trend, which is attributed to initially higher groundwater levels"; was it already decreasing during the spinup? In this case, the spinup time of 1 year may not have been sufficient. What spinup times are used in similar studies?**

   To avoid confusion, the sentence "Subsurface runoff shows a decreasing trend, which is attributed to initially higher groundwater levels "will be removed in the revised version. The observed decline in subsurface runoff over the Sahara is not a spin-up effect, as such systematic decline is not observed over other locations. Furthermore, spinup was increased to 2 years in the validation runs. Nevertheless, we will consider performing tests with a longer spinup to study the impact on the results and in case larger impacts are found, redo the simulations.

40. **L443: Again, I am confused that WFDE5 would have an hourly temporal resolution while it is states in the Methodology that it was aggregated to 3h.**

   WFDE5 has hourly temporal resolution in the initial run but 3 hourly only in the validation run.

41. **Figure 8: can we conclude from this that one shouldn't use mean soil texture properties, as it is the only one deviating from all others? Of course, this claim would be stronger if an external product were used for validation.**

   A definitive statement will be made in the revised version of the manuscript after comparison with a reference dataset.

42. **L465: "... variations in ET values across different soil texture maps", not really: only a single one shows a variation (and this is probably also the limitation of your average margin metric, which isn't able to identify this "outlier"). The following paragraph fails to mention or discuss this fact.**

   The statement will be changed to "... exhibits a variation in ET values across different soil texture maps", in the revised version.

43. **L481: "We selected one location for each of the eight climate regions"; only two are in the main text.**

   The text will be changed to "We selected one location for each of the eight climate regions. Two are presented here while the reader is referred to the supplementary materials for others."

44. **L490-492: Add a paragraph on the crop model/irrigation model in CLM5 to the Methodology.**

   A brief description of the irrigation model of CLM5 will be added to section 2.1 under CLM5.0

45. **L494-495: Not clear from the Figure.**

   The differences are not expressed in absolute terms (mm) but in terms of the percentage of ET for each forcing.

46. **L499: "While slight variations in surface runoff are found ..."; again, be more specific which soil texture maps differ, since they don't all differ from one another!**

   The statement "while slight variations in surface runoff are found for WFDE5" will be changed to reflect a quantity in the revised version.

47. **Figure 9:**

   o **Panel "S.Runoff-WFDE5": is the Mean on top of the other textures, or the Rand?**

Other textures are 0 while the mean stands out. This is further clarified in Figure S31 and Table T10 with average margin calculation.

- ○ **What is the FAO/Dom soil texture type for this grid cell?**

  Sandy loam. This will be clarified.

- ○ **The Rand soil texture may be very different when the experiment is performed again, no? Since this is only a single pixel. Hence, the study is actually not reproducible in this sense. Also see Fig. 10: here Rand just happens to coincide with Dom in terms of soil texture class, so the lines overlap. But run the experiment again and the results may be completely different. I suggest performing "Rand" several times and reporting an ensemble average to resolve this issue.**

  Running these experiments multiple times in a Monte Carlo fashion is not feasible given the required compute time for the high spatial resolution model runs over Africa. Instead, the focus of this work is to identify and not to quantify the sensitivity of uncertain soil properties to atmospheric forcing's high temporal resolution.

48. **L519-529: Same remark as earlier: the difference in soil texture maps is always just cause by Mean vs. all the rest.**

   A definite claim will be made after comparison with other datasets.

49. **Table 3: Not clear what is presented here: difference between which two things? Units?**

   After aggregating WFDE5 to 3 hourly, average margin was calculated for the ET, surface runoff, subsurface runoff and soil moisture. The table shows that WFDE5 which was aggregated to 3 hours and GWSP (3 hourly) had similar values for the margins, while WFDE5 (hourly) had larger margins. All values are in mm/month.

50. **Table 5:**

   - ○ **How come the SGMean column has a specific soil texture class? I thought this approach created a "new" class with averaged soil metrics.**

     The averaging was performed at the level of the clay and sand fractions which are in percentages. The average clay, sand and silt percentages were used to determine the soil texture class according to the USDA triangle.

   - ○ **Is referenced nowhere in the text. Please use it to discuss for example the bottom row of Fig. 9. Overall, the Discussion describes the figures**

**and results too much, rather than explaining what we see in them (and why!).**

This will be considered in the revised version of the manuscript.

51. **Conclusion: misses outlook discussing how future research can benefit from the results of the study.**

We will discuss how future research can benefit from the result of this study in the revised version of the manuscript.

**Technical corrections**

- **L55: "0.5o" instead of "0.5°"**

- **L162: "kg/m$^2$s" instead of "kg/(m$^2$s)"**

Both adjustments will be made in the revised version of the manuscript.

---

## Author Comment (AC2)

**Review Comments (egusphere-2023-3132)**

Oloruntoba et al. analysed how various soil data and forcings impact CLM5 water-related variable simulations across Africa. This work could be valuable in guiding future simulations. However, the paper is not well-written. Below, I list some of my comments for improvement.

1. The title is confusing. What do you mean by "temporal model resolution"? It needs to be modified. It does not reflect the important points mentioned in your abstract about different soil textures and forcing sources, which are crucial to your paper.

   The title will be changed to "High Resolution Land Surface Modelling over Africa: the role of uncertain soil properties in combination with forcing temporal resolution" in the revised version.

2. The abstract is not well organized. From your summary of your experiments (4 soil textures and 3 forcings), readers might expect results on how soil impacts the simulations and how different forcings affect the simulations. However, the abstract does not mention how different forcings affect your simulations, i.e., the IV point in your conclusion section. Thus, I think the abstract is not well organized.

   We wanted to focus the abstract on the main novel results, which in our opinion are more related to the combination of simulations with different soil input data and the temporal resolution of the atmospheric forcing input. We still think that this needs to be the focus, but will dedicate an additional line to the impact of different atmospheric forcings as well.

3. The introduction should be reorganized or rewritten. Many paragraphs in the introduction belong in the method section. For example, paragraphs 4 through 6, about specific soil data, forcing, and experiment design, respectively, should be mentioned in the methodology section. Therefore, the introduction should be more thoughtful and include more logical content with citations. As you have written the method in the introduction section, it makes some information in the method section somewhat fragmented; for example, L275 mentions "The four (4) soil texture maps", which originally comes from L81 of the introduction.

   Thank you for this recommendation. We will move some of the material of the introduction section to the methodology section to better explain the experimental details and ensure consistency throughout the manuscript. Also, we will modify the references in the methodology to avoid any fragmented or recurring information and to ascertain that all technical details are clearly introduced and appropriately cited.

4. In L265, please clarify if one year of spin-up is enough. Running the entire domain for a long time may be challenging, but you need to check if one year is sufficient. I suggest picking several typical grid cells with typical plant functional types and/or soil textures, to test if the one-year spin-up is enough for your analysed variables (ET, runoff) to become stable enough

to show the difference between your 12 different experiments. That is to say, can your conclusion made here represent results if run the simulations for 30 years?

Earlier works over the Southern Africa region including (Crétat et al., 2012), (Ratna et al., 2014) and (Zhang et al., 2023) have employed 6 months or less spin-up times (but they used different land surface models). For the validation runs performed in this work, spin up was also increased to two years. Nevertheless, we will perform tests with a longer spinup to study the impact on the results and in case larger impacts are found, redo the simulations.

5. For the different soil texture upscale data, is it common in our community to "randomly select a single SoilGrid cell"? If so, please provide some references; if not, why do you want to test it here if few uses it in the community?

Although, the "Random upscaling" method has not been documented, it is a novelty of this paper. Unlike other upscaling methods, the random upscaling method, creates an opportunity for outliers to be the effective soil parameter. This is because the Dominant upscaling method introduces bias towards the prevalent soil types and the Averaging method presents a conservative estimate that misses extremes. Although the random upscaling method selects for certain grid cells outlier, overall it samples correctly the pdf of texture values, which is not guaranteed by the other methods mentioned before. Since this work hinges on detecting uncertainties in soil texture information, using an un-biased upscaling method is therefore sacrosanct. Hence the consideration of the random upscaling method.

6. L274. Is the metric termed "average margin" developed by you? If not, you need to add proper citations.

Yes, it was initiated here.

7. Figure 2 could be moved to the supplementary because it is not part of your main story but an explanation for your main story.

We will consider this suggestion in the revised version.

8. It is better to use some quantitative metrics to quantify the differences in model simulations between different experiments, instead of a qualitative way, to distinguish which factors (e.g., different soil, different forcing) are the most important.

We will reformulate the text to take this comment into account and provide information on the statistical significance of differences in simulation results (ET, surface runoff) between different scenarios if possible.

9. The analysis would be better if further compared with benchmark datasets.

A comparison has been made with GLDAS dataset and will be featured in the revised manuscript.

10. L52. Correct the typo of the degree symbol from "0.5o", and check for other similar errors throughout the document.

In Line 52, "0.5o" will be changed to "0.5°". We will check the manuscript thoroughly for similar typographical errors and correct them.

11. In the Data Availability section, it would be better to include where (URL or platform) you specifically obtained the data, e.g., SoilGrids, different forcings, etc.

The Specific URL where Soil Grids and other ancillary data is downloaded will be specified in the revised manuscript.

12. The discussion section includes numerous analyses which, although sufficient, feel oddly placed. If there are so many analyses based on new tables, why not include them in the results section (e.g., Tables and new figures)? Additionally, the discussion does not provide a broad scope that shows how your study could further connect with or guide future research.

Yes, we agree that some of the tables and figures presented in the discussion could be more relevant in the result section. The findings and their corresponding tables will be moved to the results section. Also, we will expand on how our findings connect with existing literature and outline potential directions for future research.

**References**

Crétat, J., Pohl, B., Richard, Y., & Drobinski, P. (2012). Uncertainties in simulating regional climate of Southern Africa: Sensitivity to physical parameterizations using WRF. *Climate Dynamics*, *38*(3), 613–634. https://doi.org/10.1007/s00382-011-1055-8

Doherty, C. T., Johnson, L. F., Volk, J., Mauter, M. S., Bambach, N., McElrone, A. J., Alfieri, J. G., Hipps, L. E., Prueger, J. H., Castro, S. J., Alsina, M. M., Kustas, W. P., & Melton, F. S. (2022). Effects of meteorological and land surface modeling uncertainty on errors in winegrape ET calculated with SIMS. *Irrigation Science*, *40*(4), 515–530. https://doi.org/10.1007/s00271-022-00808-9

García-García, A., Cuesta-Valero, F. J., Beltrami, H., González-Rouco, F., García-Bustamante, E., & Finnis, J. (2020). Land surface model influence on the simulated climatologies of temperature and precipitation extremes in the WRF v3.9 model over North America. *Geoscientific Model Development*, *13*(11), 5345–5366. https://doi.org/10.5194/gmd-13-5345-2020

Ratna, S. B., Ratnam, J. V., Behera, S. K., Rautenbach, C. J. deW., Ndarana, T., Takahashi, K., & Yamagata, T. (2014). Performance assessment of three convective parameterization schemes in WRF for downscaling summer rainfall over South Africa. *Climate Dynamics*, *42*(11), 2931–2953. https://doi.org/10.1007/s00382-013-1918-2

Yin, Z., Ottlé, C., Ciais, P., Guimberteau, M., Wang, X., Zhu, D., Maignan, F., Peng, S., Piao, S., Polcher, J., Zhou, F., Kim, H., & other China-Trend-Stream project members. (2018). Evaluation of ORCHIDEE-MICT-simulated soil moisture over China and impacts of

different atmospheric forcing data. *Hydrology and Earth System Sciences*, *22*(10), 5463–5484. https://doi.org/10.5194/hess-22-5463-2018

Zhang, Z., Laux, P., Baade, J., Arnault, J., Wei, J., Wang, X., Liu, Y., Schmullius, C., & Kunstmann, H. (2023). Impact of alternative soil data sources on the uncertainties in simulated land-atmosphere interactions. *Agricultural and Forest Meteorology*, *339*, 109565. https://doi.org/10.1016/j.agrformet.2023.109565

---

## Author Comment (AC3)

This research conducted a comprehensive study by integrating two distinct soil mapping approaches, FAO and SoilGrids, into CLM modeling in Africa. Additionally, it investigated the impact of different soil texture upscaling methods.

The findings reveal that the origin of soil texture exerts a significant influence on simulated ET, runoff, and soil moisture content, surpassing the impact of the chosen upscaling method. This phenomenon is particularly pronounced as variations in soil properties directly affect surface water distribution. Our prior publication, which integrated soil texture into fully coupled modeling, corroborates this observation and underscores potential feedback loops between soil uncertainty and atmospheric dynamics in the African region.

Zhang, Z., Laux, P., Baade, J., Arnault, J., Wei, J., Wang, X., Liu, Y., Schmullius, C., & Kunstmann, H. (2023). Impact of alternative soil data sources on the uncertainties in simulated land-atmosphere interactions. Agricultural and Forest Meteorology, 339(March), 109565. https://doi.org/10.1016/j.agrformet.2023.109565

All these studies underscore the importance of accurately representing soil texture within land surface models, as well as in coupled land-atmosphere and earth system models. It highlights the necessity for further research to refine these soil representations for improved modeling skills.

Dear Reviewer,

Thank you for reviewing our work. We appreciate your effort and your interest in ensuring that science is at its best in this community.

We are glad that your study corroborates our observations and will consider citing your work in the revised edition of our manuscript.

Thanks once again.

Best regards,
Bamidele Oloruntoba on behalf of all co-authors.

---

## Author Response (AR1)

**Reviewer 1**

Reviewer's comment in black (bold), author's response in non-bold blue text.

This study uses the CLM5 model over Africa with different atmospheric forcings and soil texture inputs and explores their role in estimates of ET, runoff and soil moisture. Overall, there is a larger effect from the forcing dataset than from the soil texture inputs.

**General comments**

In my view, the paper in its current form requires major revisions before it can be considered for publication. The authors should specifically address the following limitations:

- **Since the study period is relatively short (two years): how can we know the results are generalizable to other years? Would it be an option to extend the study period?**

  This work investigates the interaction between uncertain soil properties and the temporal resolution of atmospheric forcings at the continental, regional and local scale with a focus on the rainfall intensity representation in the land surface model input. It therefore suggests that as long as the CLM5 model does not handle precipitation differently, extending the study period may not change the current outcome. It is also beyond the scope of this study to extend the simulation period at this high spatial resolution to a longer period.

- **The study misses an evaluation with reference data of ET, soil moisture, and runoff to better appreciate the impact of the work: which forcing, and soil texture input combination do the authors recommend the community to use in future work? Section 3.3 (local results) could be dropped in favour of this.**

  Performing an evaluation with measurement data is beyond the scope of this paper as stated in lines 67-69. The aim of the paper is not to select the best model settings over Africa or extensively evaluate the model performance, but to investigate the impact of certain model inputs (different soil input data versus different atmospheric forcings and role of temporal resolution) in a more generic sense. However, we present a comparison of our ET, runoff and soil moisture estimates with GLDAS-2.1 dataset introduced in L234-242.

[revised manuscript text omitted]

**Often small impacts are extensively discussed, but it's not clear whether they are significant or not. Statistical tests should be performed to verify this.**

Statistical significance tests have been performed and p-values are now indicated in the timeseries plots for regional and local analysis. These can be found in sections 3.2 and 3.3. In addition, text was reformulated to focus less on small impacts.

- **I have methodological concerns about using the "random selection" upscaling technique for a single grid cell and then comparing the results with other upscaling techniques. Such results are not reproducible, so how should they be interpreted?**

Random results are reproducible using "seeding" in programming. Seeding is used to initialize the random number generator (RNG) such that it makes a reproducible sequence of numbers even on different computers. We specified seeds before performing upscaling using the Random method. If we therefore perform upscaling using the Random method multiple times, we will have the same outcome.

In addition, although a random value might locally give quite arbitrary outcomes with soil properties which could deviate for that grid cell quite a lot from the mean or median value, over many grid cells this is not the case anymore and the

mean/median are reproduced by the random selection. The advantage of the random selection is that the full Probability Density Function (PDF) of soil properties at the large scale is better reproduced. This is now also better clarified in the paper, see L174-185:

"(iii)    Random selection of a single SoilGrid cell and use of the soil texture values for this grid cell for the complete 3km x 3km CLM model grid cell. This method which is a novelty of this work, creates a chance for texture outliers to define the soil hydraulic parameters. This ensures that over larger regions the Probability Density Function (PDF) of soil properties is better reproduced by the model than by selecting the dominant soil texture or average soil texture. It differs from other upscaling methods as it avoids spatial averaging or smoothing. Although it can introduce larger local biases in the soil hydraulic parameters and thus model output variables, it is not expected to induce systematic biases at larger scales, as local biases for some grid cells will be cancelled out by biases at other grid cells. In addition, as soil texture is not averaged or smoothed before processing it through the non-linear simulation model, it is expected that also model output variables, averaged over larger areas, are unbiased. We also specified a random number generator (RNG) seed which makes the randomisation reproducible in other machines. "

- **The Methodology is too detailed in terms of equations that are used by the LSM. These equations are only useful if they are referenced to in the text to explain or discuss the results, which is currently not the case.**

- Equations have been reduced in the revised manuscript and readers are now referred to the articles where the equations are discussed in detail. This is to ensure that the main text remains focused on the results, interpretation and discussion of the findings, while still providing access to the technical details for readers who require them.

- **The Discussion is too descriptive and fails to convincingly explain some of the results.**

  To ensure better clarity, we have partitioned our discussion into different sections (3.5.1 – 3.5.3) and provided further details. The sections include:

  - The role of temporal resolution in rainfall intensity representation (3.5.1)
  - The role of soil texture in water balance components (3.5.2)
  - Implications for land surface modelling and community impact (3.5.3)

  "**3.5.1 The role of temporal resolution in rainfall intensity representation**

[revised manuscript text omitted]

**Specific comments**

1. **The title hints an impact of "temporal model resolution", but this should be the resolution of the atmospheric forcing dataset (input for the model). Temporal model resolution is always 30 minutes, and its impact was not examined.**

   Yes, the land surface model temporal resolution is 30 minutes. We have revised the title in the revised manuscript version for consistency. The title has been changed to "High Resolution Land Surface Modelling over Africa: the role of uncertain soil properties in combination with forcing temporal resolution."

2. **L62-69: Part of this paragraph should go to the Methodology, for example, how the dataset was upscaled to the model resolution.**

   The paragraph has been moved and explained in detail in section 2.3.

3. **L70: many new acronyms which should be written in full the first time they are introduced.**

   CRUNCEP (Climatic Research Unit (CRU) and National Centres for Environmental Prediction (NCEP)), GSWP (Global Soil Wetness Project) and WFDE5 (Watch Forcing Data methodology on ERA 5) are now written in full in the revised version of the manuscript.

4. **L83: "the novelty of the work lies in ..."; this is more or less a repetition of line 58.**

   We agree that the two statements look similar but are left untouched as they both serve different purposes. The first tells what to expect in the paper, the second introduces the novelty of the paper.

5. **L88: the Introduction is finished with raising two main research questions on which the authors want to find the answer. I expect to explicitly find an answer to question 1 and question 2 somewhere in the Discussion or (preferably) the Conclusions.**

   Answers to the research questions are now explicitly stated as the most important findings in the study in the conclusion of the revised version of the manuscript (740-761).

   „1. Community Land Model version 5 (CLM5) model runs over the African continent were performed at a high spatial resolution of approximately 3km, with four different soil texture maps and three different atmospheric forcings. The four different soil texture inputs included the FAO soil map and three differently upscaled SoilGrids250 maps. The three different atmospheric forcings were CRUNCEPv7, GSWP3 and WFDE5. The most important findings were: Average evapotranspiration and surface runoff simulated by CLM5 over the African continent show a limited sensitivity to variations in the soil texture input. The source of soil texture information (FAO versus SoilGrids) results only in minor variations in the continental average ET or surface runoff (0.3% variations around mean), and the impact of different upscaling approaches of soil texture information is even smaller. This sensitivity to soil texture input is much smaller than the sensitivity to the different atmospheric forcings (3% variations for mean ET and 26% for surface runoff). Average subsurface runoff and average soil moisture at the continental scale are both as sensitive to variations in atmospheric forcings as to variations in soil texture information.

   2. Although average surface runoff at the continental scale shows a limited sensitivity to soil texture input, at the regional and, especially, the local scale this sensitivity is much higher, but mainly in combination with the higher temporal resolution of WFDE5 forcings (hourly). The higher temporal resolution of WFDE5 forcings (hourly) than the other atmospheric forcings resulted not only in larger variations in simulated surface runoff, but also ET and subsurface runoff for the different soil texture maps. This points to the fact that the impact of soil texture becomes more important in combination with higher temporal resolution of atmospheric forcings. We explain this with the impact of the temporal resolution of atmospheric forcings on the rainfall intensity and the partitioning of rainfall into surface runoff, which is also determined by the hydraulic conductivity of the soil.

This, in turn, affects also the amount of water available for evapotranspiration and drainage."

**L94: Add a citation to CLM5.0. Additionally, be consistent in the use of CLM5.0 and CLM5: both are currently found throughout the text.**

CLM5.0 (Community Land Model version 5.0) has been written in full in L64 and thereafter referred to as CLM5. All other uses of "CLM5.0" within the text are now changed to CLM5 in the revised version.

6. **L101: "CLM5.0 therefore has features of great interest for land surface modelling over Africa at a high resolution". Why specifically Africa?**

   The statement has been changed to (L94-L96): "Considering Africa's land surface heterogeneity, CLM5 has features of great interest for land surface modelling over Africa at a high spatial resolution."

7. **L106: Acronym for CLM5 should be written out in full the first time it is used (in the Introduction).**

   This has been addressed in our response to comment 5.

8. **L107: Change "The total porosity is given by" to "The total porosity for level i is given by".**

   Readers are now referred to the articles containing the equations for more details.

9. **L117: Citation for the Brooks and Corey model? What is the physical interpretation of "the exponent B"?**

   The citation for the Brooks and Corey model has been included (L104-109) and readers are now referred to the articles containing the equations for more details.

10. **Overall, the introduction has too many equations which are not referenced anywhere in the text. So it is not clear what their added value is. Either they should be used in the Discussion to support in explaining the observed results, or they should be replaced by a citation of the CLM5 paper in which they can probably also be found?**

    Many of the equations have now been replaced by a citation of the CLM5 paper and technical note.

11. **L162: Add that the time step Δt is expressed in seconds.**

    This is now done in L124. Readers are alsoreferred to the article containing the equations for more details.

12. **L167: What is the native resolution of IGBP-DIS? Is it 3km? If not, why is only SoilGrids250m upscaled?**

The IGBP-DIS soil texture information is 8km resolution. This was internally downscaled by CLM5. This is now mentioned in L140-143 of the revised manuscript.

13. **L170: "The 10 upper CLM5 soil layers": how many are there in total? This type of information on the model should be in section 2.1, rather than the equations which are currently there.**

There are 20 active soil layers in CLM5 (8.6m deep). In this work we are concerned with the top 2m. This is now stated in L142-144. "The soil texture dataset which is at approximately 8km resolution provides information for the top 10 CLM5 soil layers: at 0.0175, 0.0451, 0.0906, 0.1656, 0.2892, 0.493, 0.829, 1.3829, 2.2962 and 3.4332 meter depth."

14. **L181-182: Too detailed and not relevant for this study.**

The sentence has been deleted.

15. **L200: Is method (iii) the novelty of this paper? If yes, it should be mentioned explicitly and motivated (why is a third new method necessary if two other already exist). If no, a citation to another study using this approach is missing.**

Yes, the "Random upscaling" method is a novelty of this paper as it is yet to be documented. In the revised manuscript, Lines 175-185 contains:

" This method which is a novelty of this work, creates a chance for texture outliers to define the soil hydraulic parameters. This ensures that over larger regions the Probability Density Function (PDF) of soil properties is better reproduced by the model than by selecting the dominant soil texture or average soil texture. It differs from other upscaling methods as it avoids spatial averaging or smoothing. Although it can introduce larger local biases in the soil hydraulic parameters and thus model output variables, it is not expected to induce systematic biases at larger scales, as local biases for some grid cells will be cancelled out by biases at other grid cells. In addition, as soil texture is not averaged or smoothed before processing it through the non-linear simulation model, it is expected that also model output variables, averaged over larger areas, are unbiased. We also specified a random number generator (RNG) seed which makes the randomisation reproducible in other machines."

14. **Table 1: the longitudinal extent is not relevant for this study: all three are global and thus cover the study domain.**

The longitudinal extent has been removed in the revised version.

15. **L256: "A spatially varying soil thickness dataset": which one?**

The spatially varying soil thickness dataset is new in CLM5. It replaces an assumed soil thickness of 2m present in earlier versions of the CLM. The reference to the dataset has been provided in L258-259.

16. **L270: "the hourly WFDE5 forcings were also aggregated to 3 hours and used in new simulations". Perhaps this should go to section 2.4. Why not 6h, so all three forcings are comparable?**

We have now made new simulations including aggregation of WFDE5 forcing to 6 hours as well. This makes WFDE5 not just comparable to GSWP which is 3hourly but with CRUNCEP as well which is 6 hourly.

The revised manuscript has now been adjusted from L286-291.

"To further substantiate the role of soil texture input to CLM5, a new set of simulations was conducted. To ensure comparability with CRUNCEP (6 hourly) and **GSWP** (3 hourly), the hourly WFDE5 forcings were aggregated to 6 hours and 3 hours, respectively. The model was then run with the soil texture information. This was conducted to identify discrepancies between the simulation outcomes of WFDE5 at hourly, 3-hour and 6-hour temporal resolutions. Furthermore, a comparison was made with the results obtained by CRUNCEP and GSWP. The results were also analysed at the monthly level, in addition to the regional and local time series."

The results are now contained in section 3.4.3 from L650-671.

17. **L276: "A simulated variable for a certain atmospheric forcings- soil texture map combination at a given time step is denoted by M1(t), M2(t), M3(t) and M4(t)." Don't you have 12 different of such combinations? Or do you mean that you have 4 such combinations for each atmospheric forcing? Please be more specific in the definition. Also, define t as the time step.**

Yes, we have a combination of the 4 soil texture map outputs for each atmospheric forcing.

The statement has been revised in L295-302 to read:

"The average margin for a simulated variable for a certain atmospheric forcing/soil texture map combination at a given time step is denoted by M1(t), M2(t), M3(t) and M4(t)."

L302 now contains:

"where T represents the total number of time steps in the time series and t denotes time step."

18. **L290: ET is evapotranspiration, not evaporation.**

L399 now states "3.2.1 Evapotranspiration" in the revised version.

19. **L298: "marginally" implies that you performed a statistical test. One should actually be performed to assess the significance of the differences in mean/max ET (and other variables), depending on the texture, forcing and their interaction.**

Significance testing has now been performed using ANOVA tests introduced in (L303-315).

"A one-way analysis of variance (ANOVA) was conducted to ascertain whether the outputs of the four soil maps for each atmospheric forcing group exhibited significant variation. Firstly, the mean of the four soil map outputs was calculated, and the deviation of each map's output from the mean was obtained. The resulting deviations were subsequently expressed as percentages relative to the mean output, thus providing a normalised measure of the deviation for each soil map, which could then be compared with results for other atmospheric forcings. The data were subsequently transformed into a long format suitable for ANOVA, in which the percentage deviations for each soil map were compared. The dependent variables were the obtained percentage deviations, while the independent variables were the categorical variable defining the compared groups (FAO, dominant, mean and random). Subsequently, an analysis of variance (ANOVA) was conducted to ascertain whether there were statistically significant discrepancies between the models' percentage deviations. The results of the ANOVA analysis yielded a p-value statistic, which was used to determine the significance of the observed variations in soil texture map outputs at the 95% confidence interval. For further details on the ANOVA framework, we direct the reader to the works of Fisher (1925) and Brandt (2014). "

Also, the word "marginally" has been changed to "slightly" in L407.

20. **L330-332: again, the statements made here require hypothesis testing.**

The word "significantly" in the old sentence has been changed to "substantial" in L441 in a new sentence in the revised version. Information on significance tests is now available in all regional and local time series plots.

21. **L335: WFDE5 are not hourly because they have been aggregated to 3h (L270)?**

This has been properly clarified on L286-292.

"To further substantiate the role of soil texture input to CLM5, a new set of simulations was conducted. To ensure comparability with CRUNCEP (6 hourly) and GSWP (3 hourly), the hourly WFDE5 forcings were aggregated to 6 hours and 3 hours, respectively. The model was then run with the soil texture information. This was conducted to identify discrepancies between the simulation outcomes of WFDE5 at hourly, 3-hour and 6-hour temporal resolutions. Furthermore, a comparison was made with the results obtained by CRUNCEP and GSWP. The

results were also analysed at the monthly level, in addition to the regional and local time series."

22. **L337: "The increased surface runoff in the WFDE5 forced simulations reduces the availability of water for ET processes"; can this claim be made with certainty if it also rains more, so more water is coming in (cfr. Fig. 2)?**

The statement has been changed in L443-444 to:

"The increased surface runoff in the WFDE5 forced simulations reduces the availability of water for ET processes especially during sheet flow."

23. **L339: "A higher temporal resolution of the atmospheric forcings as for WFDE5 will result in higher peaks of precipitation intensity, whereas a coarser resolution of 6 hours like for CRUNCEP will average out intensive precipitation over longer time periods with less high peaks in precipitation intensity"; but wasn't this effect to be mitigated by aggregating the hourly dataset (L270)?**

This effect was mitigated in the validation set-up where WFDE5 was aggregated to 3 hourly and 6 hourly. But in the initial run, which first indicated the finding, the statement holds. This has been clarified on L286-292:

"To further substantiate the role of soil texture input to CLM5, a new set of simulations was conducted. To ensure comparability with CRUNCEP (6 hourly) and GSWP (3 hourly), the hourly WFDE5 forcings were aggregated to 6 hours and 3 hours, respectively. The model was then run with the soil texture information. This was conducted to identify discrepancies between the simulation outcomes of WFDE5 at hourly, 3-hour and 6-hour temporal resolutions. Furthermore, a comparison was made with the results obtained by CRUNCEP and GSWP. The results were also analysed at the monthly level, in addition to the regional and local time series"

24. **L353: "significantly influenced": be careful using this word.**

The statement "The estimation of subsurface runoff is more significantly influenced by soil texture variations and the upscaling of soil texture properties compared to ET and surface runoff simulations." has been changed in L459-460 to:

"The estimation of subsurface runoff is more influenced by soil texture variations and the upscaling of soil texture properties compared to ET and surface runoff simulations."

25. **L355: "The most substantial differences in simulated subsurface runoff are observed between the FAO soil map the SoilGrids250m maps"; actually the forcings still result in the largest differences.**

We have changed this sentence to (L461-L463):

"The most substantial differences in simulated subsurface runoff among soil texture inputs are observed between the FAO soil map and the SoilGrids250m maps, while the disparities among the upscaled SoilGrids250m maps are smaller especially with GSWP and CRUNCEP forcings"

26. L364: Fig. 4 should be referenced to at the start of the section.

This has been changed as suggested.

27. L371: It's confusing to talk about ET in the section on runoff, save this for the Discussion.

We have removed the sentence.

28. L382: "weighted average": how are the weights defined? Perhaps this is something to be described in the Methodology.

The weights are defined according to the thickness of each soil layer. In L267, the sentence "For continental scale results, yearly sum of evapotranspiration (ET), surface runoff, and subsurface runoff were computed as well as seasonal mean soil moisture contents." has been changed to (L282-L285):

"For continental scale results, annual mean of evapotranspiration (ET), surface runoff, and subsurface runoff were computed as well as the seasonal mean of the weighted average of the top 2 meters soil moisture content. The weights for calculating weighted average of soil moisture content were defined according to the thickness of each soil layer in CLM5 ."

29. L394: This statement doesn't hold for the mean: similar effect when varying the forcing and the texture.

The statement has been changed (L503-504):

"Like for ET and surface runoff, varying the atmospheric forcing impacted continental maximum of soil moisture content more than variations in soil texture input."

30. Figure 5:

   o **Would it make sense to also look at min soil moisture, next to mean and max?**

   Minimum values are now displayed in the updated difference maps (Figures S2-S5).

   o **Units of the colorbar are $mm^3/mm^3$, while elsewhere in the text $cm^3/cm^3$ is used. Be consistent. Also, consider using more sensible intervals for the colorbar ticks.**

Colorbar units for soil moisture are now harmonised to cm$^3$/cm$^3$ and intervals made more relatable to have binned colour classes in the new plots.

- **Perhaps one could keep the first column as is, and the other columns could show the difference w.r.t. the first column? Now all 12 maps look identical and it's hard to see where the differences are.**

  This has been done as suggested by the reviewer. New difference maps have been made and referred to in the revised manuscript (Figures S2-S5).

31. **Section 3.2: Consider moving the definitions of the regions to the Methodology.**

    The definition of regions has now been moved to section 2.6.

32. **L410: Motivate the choice for Sahara and Central Africa (dry and wet region?).**

    Yes. The choice of the Sahara and Central Africa are based on their moisture availability contrast. Line 515 now contains "We present results for two regions (Sahara and Central Africa) based on their moisture availability contrast."

33. **L417: This statement is true on average (it may well exceed this in certain pixels).**

    The statement has been changed to (L517):

    "The Sahara region is generally on average the most moisture deficient region in Africa."

34. **Figure 7:**

    - **"Soil water content", "soil moisture content", "soil moisture" are all used interchangeably. Be consistent.**

      We now adhere to using soil moisture content in the revised version.

    - **Dominant and Random are very similar, as can be expected. The first one has already been used in literature. Please motivate (in the Methodology) the advantage of Random to justify it as a new technique (e.g., are there computational gains?). Also: which one is best? Why was no evaluation performed? There are many global grid-based ET products available that can serve as reference.**

      The choice of random upscaling has now been motivated in the methodology, L175-185.

"Random selection of a single SoilGrid cell and use of the soil texture values for this grid cell for the complete 3km x 3km CLM model grid cell. This method which is a novelty of this work, creates a chance for texture outliers to define the soil hydraulic parameters. This ensures that over larger regions the Probability Density Function (PDF) of soil properties is better reproduced by the model than by selecting the dominant soil texture or average soil texture. It differs from other upscaling methods as it avoids spatial averaging or smoothing. Although it can introduce larger local biases in the soil hydraulic parameters and thus model output variables, it is not expected to induce systematic biases at larger scales, as local biases for some grid cells will be cancelled out by biases at other grid cells. In addition, as soil texture is not averaged or smoothed before processing it through the non-linear simulation model, it is expected that also model output variables, averaged over larger areas, are unbiased. We also specified a random number generator (RNG) seed which makes the randomisation reproducible in other machines."

The full comparison results using correlation and RMSE metrics for ET, Surface runoff and Soil moisture content are provided in section 3.1 on L336-379.

35. **L430: Why is the long name for ET suddenly introduced here?**

This has been corrected in the revised version.

36. **L434: "Subsurface runoff shows a decreasing trend, which is attributed to initially higher groundwater levels"; was it already decreasing during the spinup? In this case, the spinup time of 1 year may not have been sufficient. What spinup times are used in similar studies?**

This comment by the reviewer is addressed in the modified text on lines 267-279 of the paper:

"Simulation period was from the 1st of January 2011 to the 31st of December 2014 and results for the first two years were discarded (spin up years). Earlier works over the Southern Africa region including Crétat et al. (2012), Ratna et al. (2014) and Zhang et al. (2023) have employed 6 months or less spin-up time using different land surface models while Zheng et al. (2017) employed 1 year for spin-up with a predecessor of CLM5 over the Tibetan Plateau. We compared the simulated water balance components in this work with water balance components (evapotranspiration, surface runoff and soil water content) from a fresh simulation which had 11-years of spin up time and the results do not alter our initial conclusion in this study (S54-S56). Moreover, we evaluated the adequacy of the reference period employed in this study. The continental annual average of the deepest soil moisture layer was calculated, a trend line was fixed, and the statistical significance was calculated to determine whether the slope of the trend

differed significantly from zero. The resulting p-value of 0.353 indicated that the trend in soil moisture over the three-year period was not statistically significant different from zero based on a 95% confidence interval (S57), suggesting that extending the study period will not alter the current outcome."

37. **L443: Again, I am confused that WFDE5 would have an hourly temporal resolution while it is states in the Methodology that it was aggregated to 3h.**

WFDE5 has hourly temporal resolution in the initial run but 3 hourly and 6 hourly only in the validation run. It is now addressed in L286-292.

"To further substantiate the role of soil texture input to CLM5, a new set of simulations was conducted. To ensure comparability with CRUNCEP (6 hourly) and GSWP (3 hourly), the hourly WFDE5 forcings were aggregated to 6 hours and 3 hours, respectively. The model was then run with the soil texture information. This was conducted to identify discrepancies between the simulation outcomes of WFDE5 at hourly, 3-hour and 6-hour temporal resolutions. Furthermore, a comparison was made with the results obtained by CRUNCEP and GSWP. The results were also analysed at the monthly level, in addition to the regional and local time series."

**Figure 8: can we conclude from this that one shouldn't use mean soil texture properties, as it is the only one deviating from all others? Of course, this claim would be stronger if an external product were used for validation.**

A comparison has been made between the CLM5 simulated water balance components and the GLDAS-2.1 datasets. However, it must be acknowledged that the extent to which the GLDAS-2.1 dataset accurately represents reality remains uncertain.

This is indicated in L337-339:

"To assess the agreement between the CLM5-simulated water balance components and a reference dataset, a comparison was conducted with the outputs of GLDAS-2.1. We acknowledge that while the GLDAS-2.1 serves as a benchmark for comparison, the extent to which it accurately represents actual conditions in relation to CLM5 simulations remains uncertain."

For this reason, we cannot rank one soil texture information over another. Moreso, this work priortises detecting the sensitivity of CLM5 to varying soil texture information over evaluation of datasets used in the study.

38. **L465: "... variations in ET values across different soil texture maps", not really: only a single one shows a variation (and this is probably also the limitation of your average margin metric, which isn't able to identify this "outlier"). The following paragraph fails to mention or discuss this fact.**

The statement has been changed in L566-567 of the revised version to:

"Once again, we observe that only the WFDE5 atmospheric forcings exhibit a variation (not significant) in ET values across different soil texture maps, as shown in Figure 9."

This is to show that only a single forcing shows a visible variation.

39. **L481: "We selected one location for each of the eight climate regions"; only two are in the main text.**

The text has been changed in L583-588.

" We selected one location for each of the eight climate regions: Cairo (Egypt, Mediterranean), Agadez (Niger, Sahara), Abuja (Nigeria, West Africa), Addis-Ababa (Ethiopia, North-East Africa), Salong (DR Congo, Central Africa), Daar-es-Salaam (Tanzania, Central-East Africa), Windhoek (Namibia, South-West Africa) and Maseru (Lesotho, South-East Africa). Two of the eight locations are discussed due to their contrasting moisture availability while other locations are available in the supplementary materials."

40. **L490-492: Add a paragraph on the crop model/irrigation model in CLM5 to the Methodology.**

A brief description of the irrigation model of CLM5 has been added to section 2.1 under CLM5 in L127-134.

"Irrigation in CLM5 separates irrigated and rainfed crops by assigning them to separate soil columns. Irrigation is applied daily at 6am based on the difference between soil moisture content and target soil moisture taking also the crop leaf area index into account. Irrigation decisions are guided by datasets detailing areas equipped for irrigation according to Portmann et al. (2010). To constrain CLM5 irrigation, irrigation water is sourced from river storage, with provisions for supplements from ocean reserves. Alternatively, in severe cases of water scarcity, irrigation demand is dynamically adjusted to conserve river water levels. The applied irrigation in CLM5 is hard coded to bypass canopy interception, meaning it is added directly to the ground surface. More details can be found in Lawrence et al. (2018)."

41. **L494-495: Not clear from the Figure.**

This description has been removed.

42. **L499: "While slight variations in surface runoff are found ..."; again, be more specific which soil texture maps differ, since they don't all differ from one another!**

The sentence has been changed in L603-L605:

"Model simulations driven by CRUNCEP or GSWP show no variation in surface runoff as function of the soil texture map, while slight but insignificant variations in surface runoff are found for WFDE5".

43. aFigure 9:

- o **Panel "S.Runoff-WFDE5": is the Mean on top of the other textures, or the Rand?**

  Yes, Rand is on top of FAO while Mean is on top of Dominant. This is further clarified in Figure S37 and Table T10 with average margin calculation.

- o **What is the FAO/Dom soil texture type for this grid cell?**

  Sandy loam.

  L591-592 now contains the sentence:

  "The grid cell in focus is also dominated by sandy and loamy soils according to all 4 soil texture maps."

- o **The Rand soil texture may be very different when the experiment is performed again, no? Since this is only a single pixel. Hence, the study is actually not reproducible in this sense. Also see Fig. 10: here Rand just happens to coincide with Dom in terms of soil texture class, so the lines overlap. But run the experiment again and the results may be completely different. I suggest performing "Rand" several times and reporting an ensemble average to resolve this issue.**

  Running these experiments multiple times in a Monte Carlo fashion is not feasible given the required compute time for the high spatial resolution model runs over Africa. Instead, the focus of this work is to identify and not to quantify the sensitivity of uncertain soil properties to high temporal resolution of atmospheric forcings. Also, our comparison with an external dataset is a continental assessment mostly based on continental average. This reduces reliance on grid cell specific evaluation.

44. L519-529: Same remark as earlier: the difference in soil texture maps is always just cause by Mean vs. all the rest.

Our continental comparison with GLDAS-2.1 shows that CLM5 is sensitive to soil texture information variation. We refer the reviewer to the comparison results in section 3.1

45. Table 3: Not clear what is presented here: difference between which two things? Units?

After aggregating WFDE5 to 3 hourly and 6 hourly, we performed extra simulations and calculated average margin of ET, surface runoff, subsurface runoff and soil

moisture. The table shows that CRUNCEP, GSWP, WFDE5 which was aggregated to both 3 hours and 6 hours (3 hourly) had narrower margins, while WFDE5 (hourly) had wider margins.

This is further explained in L660-667:

"Table 3 shows the impact of varying soil texture map inputs on different water balance component for West Africa. Simulated variables show much less variation as function of soil texture map input for CRUNCEP, GSWP, 6-hourly aggregated WFDE5 forcings and 3-hourly aggregated WFDE5 forcings compared to 1-hourly WFDE5 forcings. Similar results are found for Abuja (Table 4), where CRUNCEP, GSWP, 6H-WFDE5 and 3H-WFDE5 forcings produce variations between 0.02 and 2.0 mm/month in ET, surface runoff and subsurface runoff as function of the soil texture map, while WFDE5 produces variations between 7.4 and 9.9 mm/month among soil texture maps. A similar observation was also made for other regions (Tables S1, S2, S4, S5, S6, S7 and S8) and locations (grid cells) (Tables S9, S10, S11, S12, S13, S14, S15 and S16)."

**Table 5:**

- **How come the SGMean column has a specific soil texture class? I thought this approach created a "new" class with averaged soil metrics.**

  The averaging was performed at the level of the clay and sand fractions which are in percentages. The average clay, sand and silt percentages were used to determine the soil texture class according to the USDA triangle. So, the texture classes are still within the USDA soil texture classes.

- **Is referenced nowhere in the text. Please use it to discuss for example the bottom row of Fig. 9. Overall, the Discussion describes the figures and results too much, rather than explaining what we see in them (and why!).**

  Table T17 is now referenced in L701-708 to describe the role of temporal resolution in rainfall intensity representation. The explanation of bottom row of former Figure 9 (Agadez) can now be found in L604-608

  "Although the texture class for Dom and Mean is Loamy Sand (LS) and for FAO and Rand Sandy Loam (SL) for the grid cell under concern (Table T17), statistically significant differences in soil water content are observed among the four soil texture maps. These differences arise because, although the soil texture classes are similar, the proportions of clay, sand, and silt vary among the four maps, resulting in different hydraulic conductivities."

Former Table 5 however is now moved to supplementary materials (T17).

46. **Conclusion: misses outlook discussing how future research can benefit from the results of the study.**

We now discussed the implications of our study to the land surface modelling community in section 3.5.3.

"This work demonstrates the critical role that high-resolution soil texture information and higher temporal resolution forcing datasets play in simulating water balance components. It highlights the need to use higher resolution soil texture information in land surface model simulations to improve the capturing of grid and sub-grid scale land surface heterogeneity. It is also necessary to provide better pedotransfer functions which link soil texture and soil hydraulic parameters which ultimately control infiltration. Higher temporal resolution of atmospheric forcing (hourly) in this work has also captured water balance dynamics differently from coarse temporal resolution atmospheric forcing which indicate a need for the community to further strengthen research to improve temporal resolution of atmospheric forcings especially over Africa. There have been advances in improving spatial resolution of atmospheric forcings (Funk et al., 2015) but this work serves as an indicator that higher temporal resolution atmospheric forcings are also needed. The works of (Hersbach et al., 2020) and (Cucchi et al., 2020) must be complemented in producing higher temporal resolution of atmospheric forcings. This advancement can eliminate the need for temporal disaggregation of precipitation as done in this work. This work showed that soil texture information is important in combination with high temporal resolution of atmospheric forcings as it impacts the division of rainfall into surface runoff and infiltration. Ultimately, land surface models also need to be better tuned to correctly reproduce this division, in the context of the higher temporal resolution of atmospheric input data and higher spatial resolution of information on soil hydraulic properties.

"

L748-753 of our conclusion also provides an outlook:

"This study therefore recommends further advances in the provision of both higher temporal resolution climate datasets and higher spatial resolution soil information over Africa. With higher spatial resolution soil information, sub-grid scale land surface heterogeneity will be handled with more accuracy. Also, higher temporal resolution climate datasets at less than 1-hour timesteps will not only eliminate the need for temporal disaggregation in land surface model applications but ensure that more accurate

atmospheric variables are supplied to the land surface model at each time steps which is often 30 minutes.

This study also highlights a specific implication for the tuning of surface runoff for land surface models in two ways. Firstly, higher spatial resolution soil texture data directly determine the hydraulic conductivity and water retention properties of soils at finer spatial scales, allowing more accurate estimation of runoff and subsequently other water balance components at each grid cell. Second, higher temporal resolution atmospheric forcing captures high-intensity rainfall events that can produce more surface runoff in a short period of time, especially on soils with low hydraulic conductivity, leading to a more accurate estimate of surface runoff at each affected grid cell for each model time step."

**Technical corrections**

- **L55: "0.5o" instead of "0.5°"**

- **L162: "kg/m$^2$s" instead of "kg/(m$^2$s)"**

  Both adjustments have been made in the revised version of the manuscript.

**Reviewer 2 comments in black, author responses in blue.**

**Review Comments (egusphere-2023-3132)**

Oloruntoba et al. analysed how various soil data and forcings impact CLM5 water-related variable simulations across Africa. This work could be valuable in guiding future simulations. However, the paper is not well-written. Below, I list some of my comments for improvement.

1. The title is confusing. What do you mean by "temporal model resolution"? It needs to be modified. It does not reflect the important points mentioned in your abstract about different soil textures and forcing sources, which are crucial to your paper.

   The title has now been changed to "High Resolution Land Surface Modelling over Africa: the role of uncertain soil properties in combination with forcing temporal resolution." in the revised version.

2. The abstract is not well organized. From your summary of your experiments (4 soil textures and 3 forcings), readers might expect results on how soil impacts the simulations and how different forcings affect the simulations. However, the abstract does not mention how different forcings affect your simulations, i.e., the IV point in your conclusion section. Thus, I think the abstract is not well organized.

   We wanted to focus the abstract on the main novel results, which in our opinion are more related to the combination of simulations with different soil input data and the temporal resolution of the atmospheric forcing input. We still think that this needs to be the focus, but we have now added additional lines to show the impact of different atmospheric forcings as well (L19-L20):

   "We found that varying the atmospheric forcing influenced simulated states and fluxes by CLM5 much more than changing soil information."

3. The introduction should be reorganized or rewritten. Many paragraphs in the introduction belong in the method section. For example, paragraphs 4 through 6, about specific soil data, forcing, and experiment design, respectively, should be mentioned in the methodology section. Therefore, the introduction should be more thoughtful and include more logical content with citations. As you have written the method in the introduction section, it makes some information in the method section somewhat fragmented; for example, L275 mentions "The four (4) soil texture maps", which originally comes from L81 of the introduction.

   Paragraphs 4 and 5 are now removed from the introduction. The old paragraph 4 is now included into section 2.2 (L136-161) and paragraph 5 into section 2.4 (L187-246). Paragraph 6 (L70-78) is however left in the introduction as an overview of activities performed in this study.

Line 272:" The four (4) soil texture maps were considered each providing a unique output at every timestep." was also retained as it is part of the methodology for calculating average margin.

We believe that these adjustments make the introduction more logical and cohesive especially with the movements of paragraph 4 and 5 to the methodology section. Here are specific details:

**Paragraph reorganisation:** The original introduction's paragraphs 4 and 5, which discussed specific soil data, meteorological forcings, and experiment design, were transferred to the methodology section. This modification has resulted in a reduction of the new introduction's methodological detail.

**Structure and Focus:** The revised introduction now places more emphasis on the study's motivation and the broader challenges inherent to land surface modelling across the African continent. It begins with an explanation of existing challenges related to the heterogeneity and uncertainties of soil properties and atmospheric forcings and how these impact model performance. This context then transitions to the study's purpose, emphasizing the knowledge gaps that the research aims to address.

**Logical Flow and Citations**: The revised introduction follows a logical progression, with citations that provide detailed discussion of the concepts introduced. Each paragraph builds on the previous one to deepen the reader's understanding of the study's motivation. Key concepts, such as the influence of soil and atmospheric data on land surface modeling accuracy, are introduced alongside relevant literature to substantiate the discussion. It no longer includes specific experimental designs, which are more appropriately addressed in the methods section.

In L265, please clarify if one year of spin-up is enough. Running the entire domain for a long time may be challenging, but you need to check if one year is sufficient. I suggest picking several typical grid cells with typical plant functional types and/or soil textures, to test if the one-year spin-up is enough for your analysed variables (ET, runoff) to become stable enough to show the difference between your 12 different experiments. That is to say, can your conclusion made here represent results if run the simulations for 30 years?

The comment about sufficiency of spin-up time is addressed in the revised manuscript on lines 268-279:

"Earlier works over the Southern Africa region including Crétat et al. (2012),  Ratna et al. (2014) and Zhang et al. (2023) have employed 6 months or less spin-up times using different land surface models while Zheng et al. (2017) employed 1 year for spin-up with a predecessor of CLM5 over the Tibetan Plateau. We compared the simulated water balance components in this work with water balance components

(evapotranspiration, surface runoff and soil water content) from a fresh simulation which had 11-years of spin up time and the results do not alter our initial conclusion in this study (S54-S56). Moreover, we evaluated the adequacy of the reference period employed in this study. The continental annual average of the deepest soil moisture layer was calculated, a trend line was fixed, and the statistical significance was calculated to determine whether the slope of the trend differed significantly from zero. The resulting p-value of 0.353 indicated that the trend in soil moisture over the three-year period was not statistically significant based on a 95% confidence interval (S57), suggesting that extending the study period will not alter the current outcome."

In light of the aforementioned considerations, we conclude that the spin-up period was sufficient for ET and runoff to reach a stable state. Furthermore, we believe that the period under review was sufficient for the purposes of this study.

4. For the different soil texture upscale data, is it common in our community to "randomly select a single SoilGrid cell"? If so, please provide some references; if not, why do you want to test it here if few uses it in the community?

Our motivation for the use of random upscaling is now stated in the revised manuscript on lines 174-185:

(i) "Random selection of a single SoilGrid cell and use of the soil texture values for this grid cell for the complete 3km x 3km CLM model grid cell. This method which is a novelty of this work, creates a chance for texture outliers to define the soil hydraulic parameters. This ensures that over larger regions the Probability Density Function (PDF) of soil properties is better reproduced by the model than by selecting the dominant soil texture or average soil texture. It differs from other upscaling methods as it avoids spatial averaging or smoothing. Although it can introduce larger local biases in the soil hydraulic parameters and thus model output variables, it is not expected to induce systematic biases at larger scales, as local biases for some grid cells will be cancelled out by biases at other grid cells. In addition, as soil texture is not averaged or smoothed before processing it through the non-linear simulation model, it is expected that also model output variables, averaged over larger areas, are unbiased. We also specified a random number generator (RNG) seed which makes the randomisation reproducible in other machines. "

5. L274. Is the metric termed "average margin" developed by you? If not, you need to add proper citations.

Yes, it was suggested here. We have also described it in L295-302:

"A metric termed "average margin" was introduced to quantify the impact of the temporal resolution of the atmospheric forcings in combination with soil texture map

variation. The four (4) soil texture maps were considered each providing a unique output at every timestep within the time series. The average margin for a simulated variable for a certain atmospheric forcing/soil texture map combination at a given time step is denoted by M1(t), M2(t), M3(t) and M4(t). The difference in the maximum and minimum simulated value for the variable, between the soil texture maps at a given time step is then computed as:

$$D(t) = \max(M_1(t), M_2(t), M_3(t), M_4(t)) - \min(M_1(t), M_2(t), M_3(t), M_4(t)) \quad\quad (4)$$

and the average margin is given by:

$$A = \frac{1}{T}\sum_{t=1}^{T} D(t) \quad\quad\quad\quad (5)$$

where $T$ represents the total number of time steps in the time series and t denotes time step."

6.  Figure 2 could be moved to the supplementary because it is not part of your main story but an explanation for your main story.

    We have moved Figure 2 to supplementary. It is now Figure S9.

7.  It is better to use some quantitative metrics to quantify the differences in model simulations between different experiments, instead of a qualitative way, to distinguish which factors (e.g., different soil, different forcing) are the most important.

    Thanks, we have included significance tests using ANOVA in the revised manuscript. The ANOVA tests were introduced in line 303-315 The results of the significance tests (p-value) which show whether the differences among all soil texture maps are significant are now available in all regional and local time series plots in this study.

    "A one-way analysis of variance (ANOVA) was conducted to ascertain whether the outputs of the four soil maps for each atmospheric forcing group exhibited significant variation. Firstly, the mean of the four soil map outputs was calculated, and the deviation of each map's output from the mean was obtained. The resulting deviations were subsequently expressed as percentages relative to the mean output, thus providing a normalised measure of the deviation for each soil map, which could then be compared with results for other atmospheric forcings. The data were subsequently transformed into a long format suitable for ANOVA, in which the percentage deviations for each soil map were compared. The dependent variables were the obtained percentage deviations, while the independent variables were the categorical variable defining the compared groups (FAO, dominant, mean and random). Subsequently, an analysis of variance (ANOVA) was conducted to ascertain whether there were statistically significant discrepancies between the models' percentage deviations. The results of the ANOVA analysis yielded a p-value statistic, which was used to determine the significance of the observed variations in soil texture map outputs at the 95% confidence interval. For further details on the ANOVA framework, we direct the reader to the works of Fisher (1925) and Brandt (2014). "

8. The analysis would be better if further compared with benchmark datasets.

A comparison has been made with the GLDAS-2.1 dataset and has been featured in the revised manuscript. We introduce the GLDAS-2.1 dataset in lines 234-242.

"**GLDAS-2.1**. The Global Land Data Assimilation System was originally developed to absorb satellite- and ground-based observational data products, using advanced land surface modelling and data assimilation techniques, in order to generate premium fields of land surface states and fluxes (Rodell et al., 2004). The GLDAS-2.1 dataset, which was reprocessed in January 2020, delivers monthly 0.25-degree data produced by temporal averaging of 3-hourly simulations using the Noah Model 3.6 in LIS Version 7. The GLDAS-2.1 simulations were driven by NOAA/GDAS atmospheric fields, GPCP V1.3 precipitation data, and AGRMET radiation variables from March 2001 onward. Table 1 summarizes details regarding the different meteorological forcing datasets used in this work."

The comparison was performed over the study period per grid cell as described on lines 316-323.

"Finally, we compared the different soil texture map outcomes with GLDAS-2.1 datasets as a credible benchmark to know which soil texture map or upscaling methods provided estimates closer to an established external dataset. We compared ET, surface runoff and soil moisture content using the Pearson correlation (Pearson & Henrici, 1997) to measure the strength of relationship between the datasets, Mean Absolute Error (MAE) to describe the average magnitude of the errors and Root Mean Square Error (RMSE) to emphasize large errors. More details about both the RMSE and MAE and their proper use cases are described by Hodson (2022). For the reference study period, for every grid cell and all time steps the calculated water balance components were compared with the ones from the GLDAS-2.1 dataset. This comparison was performed on a grid cell-by-grid cell basis, resulting in a complete continental assessment of the water balance components."

The comparison results are presented in Section 3.1 (L336-395). We provide an assessment of the agreement between the CLM5-simulated water balance components and GLDAS-2.1.

"3.1. Comparison of simulated water balance components with GLDAS-2.1 Datasets

To assess the agreement between the CLM5-simulated water balance components and a reference dataset, a comparison was conducted with the outputs of GLDAS-2.1. We acknowledge that while the GLDAS-2.1 serves as a benchmark for comparison, the extent to which it accurately represents actual measurements in relation to CLM5 simulations remains uncertain.

Evapotranspiration

The correlation of CLM5 simulated ET with GLDAS (Figure 1) shows a clear spatial gradient across Africa. Strong positive correlations above 0.75 as referenced in hydrology studies over Africa (Scanlon et al., 2022; Larbi et al., 2020) are mainly seen in the equatorial region and parts of Eastern Africa, Southern Africa and Madagascar, indicating acceptable model performance in these regions. Northern Africa, some parts of Central Africa, and the cape of South Africa tend to show moderate to weak positive correlations, with some areas having negative correlation (down to around -0.79). The mean correlation values vary between 0.64 and 0.70, depending on the input of atmospheric forcings and soil properties, reflecting relatively moderate agreement with GLDAS across the continent. RMSE for ET (Figure S50) displays a concentration of lower errors in the moisture deficient Northern and Southern parts of Africa, while the moisture richer Central and Eastern regions show higher RMSE values. It is important to note however that RMSE scores are magnitude dependent as they increase or decrease with the magnitude of evaluated variables.

Surface Runoff

Surface runoff correlations (Figure 3) over Africa exhibit wide variability, with very high positive correlations (up to 1.0) in Savannah regions of West Africa including parts of Namibia, Zambia and Mozambique. There are however areas with low to strongly negative correlations, particularly in the Sahara region including countries like Mauritania, Mali, Algeria, Libya, Egypt and Sudan, where correlation values are as low as -1.0. This high variability results in an average continental correlation in the range of 0.50-0.58. The RMSE for surface runoff over Africa (Figure S52) shows minimal errors in water scarce Northern and South-western Africa, with the highest RMSE values ranging from 0-11mm/month Central Africa and Western regions show relatively higher RMSE values. The high RMSE values suggest substantial discrepancies in surface runoff simulation between CLM5 and GLDAS, especially in equatorial areas.

Soil Moisture

Soil moisture correlations with GLDAS (Figure 2) show a slightly different spatial pattern compared to ET. The highest correlations (strong positive) are generally observed above the equator, top fringes of Southern Africa and Northern Madagascar. Strong negative correlations however are found in parts of Sahara specifically Mauritania, Mali, Algeria, Egypt and Sudan where certain grid cells exhibit correlations as low as -0.79. Overall, the average correlations for soil moisture are lower than for ET with a range of 0.56-0.67, indicating less correlation across the continent compared to ET. The RMSE for soil moisture (Figure S51) is 0.05-0.06 $cm^3/cm^3$. RMSE is higher in parts of Central Africa like Congo DR, where errors peak around 0.26-0.27 $cm^3/cm^3$. This RMSE pattern suggests that the CLM5 simulated soil moisture maintains a relatively stable agreement with GLDAS having minimal extreme errors across the continent.

"

9. L52. Correct the typo of the degree symbol from "0.5o", and check for other similar errors throughout the document.

"0.5o" has been changed to "0.5°". We have also checked the manuscript thoroughly for similar typographical errors and corrected them.

10. In the Data Availability section, it would be better to include where (URL or platform) you specifically obtained the data, e.g., SoilGrids, different forcings, etc.

The Specific URL where Soil Grids and other ancillary data including the newly involved GLDAS-2.1 is downloaded has been specified in the data availability section.

11. The discussion section includes numerous analyses which, although sufficient, feel oddly placed. If there are so many analyses based on new tables, why not include them in the results section (e.g., Tables and new figures)? Additionally, the discussion does not provide a broad scope that shows how your study could further connect with or guide future research.

Yes, we agree that some of the tables and figures which were presented in the discussion are more relevant in the results section. Some tables and figures have now been moved to the results section while some others have been moved to supplementary materials.

Here are the changes made:

Table 3 and Table 4 are now in the results section while the Table showing total number of precipitation events above 3mm/hour is now Table T17 of the supplementary material.

To ensure better clarity, we have partitioned our discussion into different sections (3.5.1 – 3.5.3) and provided further details. The sections include:

[revised manuscript text omitted]

the Royal Society of London. Series A, Containing Papers of a Mathematical or

Physical Character*, *187*, 253–318. https://doi.org/10.1098/rsta.1896.0007

Ratna, S. B., Ratnam, J. V., Behera, S. K., Rautenbach, C. J. deW., Ndarana, T., Takahashi, K., & Yamagata, T. (2014). Performance assessment of three convective parameterization schemes in WRF for downscaling summer rainfall over South Africa. *Climate Dynamics*, *42*(11), 2931–2953. https://doi.org/10.1007/s00382-013-1918-2

Rodell, M., Houser, P. R., Jambor, U., Gottschalck, J., Mitchell, K., Meng, C.-J., Arsenault, K., Cosgrove, B., Radakovich, J., Bosilovich, M., Entin, J. K., Walker, J. P., Lohmann, D., & Toll, D. (2004). The Global Land Data Assimilation System. *Bulletin of the American Meteorological Society*, *85*(3), 381–394. https://doi.org/10.1175/BAMS-85-3-381

Shuai, P., Chen, X., Mital, U., Coon, E. T., & Dwivedi, D. (2022). The effects of spatial and temporal resolution of gridded meteorological forcing on watershed hydrological responses. *Hydrology and Earth System Sciences*, *26*(8), 2245–2276. https://doi.org/10.5194/hess-26-2245-2022

Zhang, Z., Laux, P., Baade, J., Arnault, J., Wei, J., Wang, X., Liu, Y., Schmullius, C., & Kunstmann, H. (2023). Impact of alternative soil data sources on the uncertainties in simulated land-atmosphere interactions. *Agricultural and Forest Meteorology*, *339*, 109565. https://doi.org/10.1016/j.agrformet.2023.109565

Zheng, D., Velde, R. V. D., Su, Z., Wen, J., & Wang, X. (2017). Assessment of Noah land surface model with various runoff parameterizations over a Tibetan river. *Journal of Geophysical Research: Atmospheres*, *122*(3), 1488–1504. https://doi.org/10.1002/2016JD025572